# EUGENE: EXPLAINABLE STRUCTURE-AWARE GRAPH EDIT DISTANCE ESTIMATION WITH GENERALIZED EDIT COSTS

## ABSTRACT

The need to identify graphs with small structural distances from a query arises in domains such as biology, chemistry, recommender systems, and social network analysis. Among several methods for measuring inter-graph distance, Graph Edit Distance (GED) is preferred for its comprehensibility, though its computation is hindered by **NP**-hardness. Optimization based heuristic methods often face challenges in providing accurate approximations. State-of-the-art GED approximations predominantly utilize neural methods, which, however: (i) lack an *explanatory* edit path corresponding to the approximated GED; (ii) require the NP-hard generation of ground-truth GEDs for training; and (iii) necessitate separate training on each dataset. In this paper, we propose EUGENE, an efficient, algebraic, and structure-aware optimization based method that estimates GED and also provides edit paths corresponding to the estimated cost. Extensive experimental evaluation demonstrates that EUGENE achieves state-of-the-art GED estimation with superior scalability across diverse datasets and generalized cost settings.

## 1 INTRODUCTION AND RELATED WORK

*Graph Edit Distance (GED)* quantifies the dissimilarity between a pair of graphs (Bai et al., 2020; Doan et al., 2021; Bai et al., 2019; Ranjan et al., 2022). It finds application in identifying the graph in a collection most similar to a query graph. Given graphs $\mathcal{G}_1$ and $\mathcal{G}_2$, GED is the minimum cost to transform $\mathcal{G}_1$ into $\mathcal{G}_2$ through *edit operations*, rendering $\mathcal{G}_1$ isomorphic to $\mathcal{G}_2$. These operations comprise the addition and deletion of edges and nodes and the replacement of their labels, each linked to a cost. Figure 1 presents an example. GED computation is **NP**-hard (Zeng et al., 2009) and **APX**-hard (Lin, 1994), hence a challenging task.

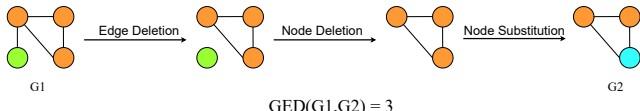

Figure 1: An *edit path* between graphs $\mathcal{G}_1$ and $\mathcal{G}_2$ with GED 3; each edit operation costs 1.

Owing to the problem's hardness, several algorithms approximate GED (Blumenthal et al., 2019a). *Optimization* based heuristic GED estimation methods employ strategies such as transformations to the linear-sum assignment problem with error correction or constraints (e.g., NODE (Justice & Hero, 2006), BRANCH-TIGHT (Blumenthal & Gamper, 2018)) and linear-programming relaxations of mixed integer programming (MIP) formulations (e.g., F1 (Lerouge et al., 2017), ADJ-IP (Justice & Hero, 2006), COMPACT-MIP (Blumenthal & Gamper, 2020)). Still, these approaches often afford only limited approximation accuracy.

Recent works have evinced that graph neural networks (GNNs) can achieve state-of-the-art accuracy in approximating GED (Jain et al., 2024; Ranjan et al., 2022; Wang et al., 2021; Bai et al., 2019; 2020; Doan et al., 2021; Li et al., 2019; Zhang et al., 2021; Piao et al., 2023). The general pipeline in this paradigm is to train a GNN-based architecture on a set of graph pairs along with their true GED distance. Some techniques also require the node mapping corresponding to the GED (Piao et al., 2023; Wang et al., 2021).

Although they afford superior accuracy, neural approaches suffer from notable drawbacks:

- **Reliance on NP-hard ground truth**: Generating training data, i.e., *true* GEDs of graph pairs, is prohibitively costly for large graphs, as GED computation is **NP**-hard. Training data are thus limited to graphs of at most 25 nodes, undermining generalizability to larger ones (§ 4).

- **Lack of interpretability**: Most of them furnish a GED between two graphs *but not* an edit path that entails it; such edit paths reveal crucial functions of protein complexes (Singh et al., 2008a), image alignment (Conte et al., 2003), and gene regulatory pathways (Chen et al., 2018). Some neural methods, e.g., GEDGNN (Piao et al., 2023) and GENN-$A^*$ (Wang et al., 2021) offer interpretability, albeit at the expense of accuracy and/or scalability, as we show in § 4.

- **Lack of generalizability**: Neural approximators do not generalize across datasets. For datasets across different domains (such as chemical compounds vs. function-call graphs), the node label set changes. As the number of parameters in a GNN is a function of the feature dimension in each node, a GNN trained on one domain cannot transfer to another, necessitating *separate* training for each dataset. As training data generation is **NP**-hard, the pipeline is resource-intensive.

In this paper, we present an optimization based algebraic method called EUGENE: Explainable Structure-aware Graph Edit Distance, which achieves state-of-the-art accuracy and is: **(1)** *optimization* based heuristic, hence does not require training; **(2)** *CPU-bound*, therefore unshackled from GPU requirements and resultant greenhouse emissions; and **(3)** *interpretable*. The innovations empowering these properties are as follows:

- **Optimization problem formulation:** We cast the GED computation problem as an optimization problem extending over Unrestricted Graph Alignment (UGA), grounded on adjacency matrices, over the space of all possible node alignments, represented via *permutation matrices*; this formulation facilitates an optimization based solution, eschewing the need for ground-truth data generation and data-specific training.

- **Interpretability:** To approximate GED, EUGENE minimizes a function over the set of *doubly stochastic* matrices, leading to a convex optimization problem that can be solved by ADAM (Kingma & Ba, 2015). We further refine the approximation by exhorting the doubly stochastic matrix using permutation inducing regularizers and inverse relabelling strategy. By operating directly on matrices, EUGENE yields a GED approximation *explainable* via a node-to-node correspondence.

- **Experimental evaluation:** Extensive experiments encompassing 15 state-of-the-art baselines over 9 datasets and 3 combinations of edits costs establish that EUGENE consistently achieves superior accuracy in GED approximation. Notably, EUGENE, does not rely on training data and thus offers a resource-efficient, GPU-free execution pipeline, which exhibits up to 30 times lower carbon emissions than its neural counterparts.

## 2 PRELIMINARIES AND PROBLEM FORMULATION

**Definition 1** (Graph). *A node-labeled undirected graph is a triple $\mathcal{G}(\mathcal{V}, \mathcal{E}, \mathcal{L})$ where $\mathcal{V} = [n] \equiv \{1, \ldots, n\}$ is the node set, $\mathcal{E} \subseteq [n] \times [n]$ is the edge set, and $\mathcal{L} : \mathcal{V} \to \Sigma$ is a labeling function that maps nodes to labels, where $\Sigma$ is the set of all labels.*

The *adjacency matrix* of $\mathcal{G}$ is $A = [a_{i,j}]_{i,j \in [n]} \in \{0, 1\}^{n \times n}$ such that $a_{ij} = a_{ji} = 1$ if and only if $(i, j) \in E$. We use $\mathbf{1}$ to denote an all-ones vector, $J$ to denote an all-ones square matrix, and $O$ to denote an all-zero square matrix.

**Definition 2** (Permutation and Doubly Stochastic Matrices). *A permutation matrix of size $n$ is a binary-valued matrix $\mathbb{P}^n = \{P \in \{0, 1\}^{n \times n} : P\mathbf{1} = \mathbf{1}, P^T\mathbf{1} = \mathbf{1}\}$. A doubly stochastic matrix of size $n$ is a real-valued matrix $\mathbb{W}^n = \{W \in [0, 1]^{n \times n} : W\mathbf{1} = \mathbf{1}, W^T\mathbf{1} = \mathbf{1}\}$.*

We define a *quasi-permutation matrix* as a matrix that is *almost* a permutation matrix.

**Definition 3** (Entry-wise norm). *Let $A = [a_{ij}]_{i,j \in [n]} \in \mathbb{R}^{n \times n}$ and $p \in \mathbb{N}^+ \cup \{\infty\}$. We define the entry-wise p-norm of $A$ as $\|A\|_p = \left( \sum_{i=1}^{n} \sum_{j=1}^{n} |a_{ij}|^p \right)^{1/p}$ for $p \in \mathbb{N}+$, and $\|A\|_\infty = \max_{i,j} |a_{i,j}|$. We denote the entry-wise 2-norm (i.e., the Frobenius norm) as $\| \cdot \|_F$.*

We denote the *trace* of a matrix $A$ as $tr(A)$.

**Definition 4** (Node mapping). *Given two graphs $\mathcal{G}_1$ and $\mathcal{G}_2$ of $n$ nodes, a node mapping between $\mathcal{G}_1$ and $\mathcal{G}_2$ is a bijection $\pi : \mathcal{V}_1 \to \mathcal{V}_2$ where $\forall v \in \mathcal{V}_1, \pi(v) \in \mathcal{V}_2$.*

Given graphs $\mathcal{G}_1$ and $\mathcal{G}_2$ with node counts $n_1$ and $n_2$, respectively, $n_1 < n_2$, we add $(n_2 - n_1)$ isolated *dummy* nodes with label $\epsilon$ to $\mathcal{G}_1$. Henceforward, we assume the two given graphs are of the same size.

**Definition 5** (Graph Edit Distance under mapping $\pi$). *GED between $\mathcal{G}_1$ and $\mathcal{G}_2$ under $\pi$ is:*

$$GED_\pi(\mathcal{G}_1, \mathcal{G}_2) \ = \ \sum_{v \in \mathcal{V}_1} d_v(\mathcal{L}(v), \mathcal{L}(\pi(v))) + \sum_{\substack{\langle v_1, v_2 \rangle \in \mathcal{V}_1 \times \mathcal{V}_1 \wedge \\ v_1 < v_2}} d_e(\langle v_1, v_2 \rangle, \langle \pi(v_1), \pi(v_2) \rangle) \quad (1)$$

*where $d_v$ and $d_e$ are distance functions over the node labels and node pairs respectively.*

The distance between two identical node labels is 0. If an existing edge is mapped to a non-existing edge, i.e, either $\langle v_1, v_2 \rangle \notin \mathcal{E}_1$ or $\langle \pi(v_1), \pi(v_2) \rangle \notin \mathcal{E}_2$ the cost[1] is $\kappa^2$, otherwise 0. Intuitively, mapping from a dummy node/edge to a real one expresses insertion, while mapping from a real node/edge to a dummy one expresses deletion, and mapping from a real node to a real node of different label denotes replacement. Figure C in the appendix illustrates GED mappings with examples.

**Definition 6** (GED). *GED is the minimum distance among all mappings.*

$$GED(\mathcal{G}_1, \mathcal{G}_2) = \min_{\forall \pi \in \Phi(\mathcal{G}_1, \mathcal{G}_2)} GED_\pi(\mathcal{G}_1, \mathcal{G}_2) \quad (2)$$

$\Phi(\mathcal{G}_1, \mathcal{G}_2)$ *denotes all possible node maps from $\mathcal{G}_1$ to $\mathcal{G}_2$.*

### 2.1 MAPPING GED TO GRAPH ALIGNMENT

We now establish that unrestricted graph alignment (UGA) (Skitsas et al., 2023) forms an instance of GED. Building on this connection, we recast GED by Definition 6 as a *generalized* graph alignment problem, leading to algebraic methods for GED estimation.

**Definition 7** (Unrestricted Graph Alignment). *Unrestricted graph alignment calls to find a bijection $\pi : \mathcal{V}_1 \to \mathcal{V}_2$ that minimizes edge disagreements between the two graphs. Formally:*

$$\min_{\pi \in \Phi(\mathcal{G}_1, \mathcal{G}_2)} \|AP_\pi - P_\pi B\|_F^2, \quad (3)$$

*Here, $A$ and $B$ are the adjacency matrices of graphs $\mathcal{G}_1$ and $\mathcal{G}_2$, respectively, $\|.\|_F$ denotes the Frobenius Norm, and $P_\pi$ is a permutation matrix , where $P_\pi[i,j] = 1$ if $\pi(i) = j$, otherwise 0.*

The proof of the following theorem is in Appendix B.

**Theorem 1.** *Given graphs $\mathcal{G}_1$ and $\mathcal{G}_2$ of size $n$, if the edge insertion and deletion cost is $\kappa^2 = 2$ and node substitution cost is 0, then $GED(\mathcal{G}_1, \mathcal{G}_2) = \min_{\pi \in \Phi(\mathcal{G}_1, \mathcal{G}_2)} \|AP_\pi - P_\pi B\|_F^2$.*

## 3 EUGENE: PROPOSED METHOD

While Theorem 1 establishes graph alignment as a special case of GED, Equation (3) assumes a specific instance of edits costs and ignores node labels, setting node edit costs to 0. We next frame GED as a generalized graph alignment problem with *arbitrary* edit costs.

### 3.1 GED AS GENERALIZED GRAPH ALIGNMENT

Given graphs $\mathcal{G}_1$ and $\mathcal{G}_2$, arbitrary costs for node edits, and cost $\kappa^2$ for edge edits, where $\kappa$ is a scalar, we propose a closed-form expression for *generalized* graph alignment:

$$\min_{\pi \in \Phi(\mathcal{G}_1, \mathcal{G}_2)} \frac{\|\tilde{A}P_\pi - P_\pi \tilde{B}\|_F^2}{2} + tr(P_\pi^T D) \quad (4)$$

Let $A, B$ be adjacency matrices of $\mathcal{G}_1, \mathcal{G}_2$, respectively, having extended the smaller graph to the size of the larger by adding dummy nodes. We set $\tilde{A} = \kappa \cdot A, \tilde{B} = \kappa \cdot B$ and define $D$ as:

$$d_{ij} = \begin{cases} d_v(\epsilon, \mathcal{L}(j)), & \text{if } i \text{ is a dummy node in } \mathcal{G}_1 \\ d_v(\mathcal{L}(i), \epsilon), & \text{if } j \text{ is a dummy node in } \mathcal{G}_2 \\ d_v(\mathcal{L}(i), \mathcal{L}(j))), & \text{if } \mathcal{L}(i) \neq \mathcal{L}(j) \end{cases} \quad (5)$$

where $d_v$ is the distance function over the node labels by Definition 5 and $\epsilon$ is the label assigned to dummy nodes. We show that, with $\tilde{A}, \tilde{B}, D$ as above, Equation (4) amounts to GED with arbitrary edit costs. Intuitively, the first term captures edge edits under mapping $\pi$, the second term node edits. The proof is in Appendix B.

---

[1]We define it to be $\kappa^2$ instead of $\kappa$ since it eases the notational burden in subsequent derivations.

**Theorem 2.** *Given two graphs $\mathcal{G}_1$ and $\mathcal{G}_2$ of size $n$, $GED(\mathcal{G}_1, \mathcal{G}_2) = \min_{\pi \in \Phi(\mathcal{G}_1, \mathcal{G}_2)} \frac{||\tilde{A}P_\pi - P_\pi \tilde{B}||_F^2}{2} + tr(P_\pi^T D)$, where $\tilde{A}$, $\tilde{B}$ and $D$ are defined as above.*

IPFP (Bougleux et al., 2017) also formulates GED as a quadratic assignment problem, yet it flattens the permutation matrix into a vector and operates on a cost matrix $C = (c_{ik,jl})_{i,k,j,l}$, where $c_{ik,jl}$ denotes the cost of editing edge $(i, j)$ in one graph to edge $(k, l)$ in the other. In contrast, EUGENE preserves the permutation matrix structure and operates on adjacency matrices $A$ and $B$, expressing edge discrepancies through the difference of the permuted matrices $\tilde{A}P$ and $P\tilde{B}$. This structure-aware formulation reduces time complexity from $O(n^4)$ in IPFP to $O(n^3)$ and is also more space-efficient: while $C$ is a dense matrix of size $n^2 \times n^2$, $\tilde{A}$ and $\tilde{B}$ are $n \times n$ and usually sparse. Moreover, EUGENE is numerically more stable, while $C$ becomes ill-conditioned and thus unsuitable for gradient-based optimization for similar $A$ and $B$, which render the rows and columns of $C$ nearly linearly dependent. Besides, EUGENE naturally accommodates permutation and doubly-stochastic constraints and maintains a spectral connection to the eigenvalues of $A$ and $B$, which enables the use of spectral techniques (Hermanns et al., 2021; Knossow et al., 2009; Singh et al., 2008b). Lastly, IPFP relies on off-the-shelf optimization methods, while EUGENE uses a custom optimization strategy, which confers the advantages shown in § 4.

Grounded in our structure-aware reformulation of GED as generalized graph alignment problem based on adjacency matrices, we can leverage advances in graph alignment for GED estimation purposes. FUGAL (Bommakanti et al., 2024), the current state-of-the-art solution for UGA, relaxes a quadratic assignment problem with an objective built on a non-convex correlation term to the feasible set of doubly stochastic matrices and applies the Frank–Wolfe algorithm (Frank & Wolfe, 1956) guided by a Sinkhorn–Knopp normalization (Cuturi, 2013) to iteratively step within that feasible set in a direction most aligned with the negative gradient. As our experimental study reveals, while this approach is good enough for graph alignment, where solutions are evaluated by the proportion of correctly aligned nodes, it yields poor results in terms of GED, where solutions are strictly evaluated by the difference of their GED cost from the ground truth. We conclude that GED estimation calls for a more rigorous approach directly targeting the convex GED cost as the core objective with stable gradient updates. Nonetheless, we adopt from FUGAL the idea of refining a doubly stochastic matrix towards a quasi-permutation matrix.

### 3.2 Permutation-Inducing Regularization

While Equation (4) provides a closed-form expression, finding the permutation matrix that minimizes it is notoriously hard, as the space of permutation matrices is not convex. To circumvent this non-tractability, we relax Equation (3) form the set of permutation matrices to that of doubly stochastic matrices $\mathbb{W}^n$, rendering the problem convex (Bento & Ioannidis, 2018), and solve the relaxed form of Equation (4):

$$\min_{P \in \mathbb{W}^n} \frac{||\tilde{A}P - P\tilde{B}||_F^2}{2} + tr(P^T D) \tag{6}$$

$$\text{Constraints: } P\mathbf{1} = \mathbf{1}, P^T\mathbf{1} = \mathbf{1}, 0 \le P_{ij} \le 1$$

Equation 6 is convex, as it minimizes a convex function over a convex domain (Boyd & Vandenberghe, 2004) and solvable with Adam (Kingma & Ba, 2015), yet the optimal doubly-stochastic matrix does not solve our exact problem. Still, these two matrix domains are connected as follows (Bommakanti et al., 2024); the proofs are in Appendix B.

**Lemma 1.** *A doubly-stochastic matrix $A$ with $tr(A^T(J - A)) = 0$ is a permutation matrix.*

Utilizing this connection, we add a bias to our objective function in the following form.

$$\min_P \frac{||\tilde{A}P - P\tilde{B}||_F^2}{2} + \mu \cdot (tr(P^T D)) + \lambda \cdot (tr(P^T(J - P))) \tag{7}$$

$$\text{Constraints: } P\mathbf{1} = P^T\mathbf{1} = \mathbf{1}, 0 \le P_{ij} \le 1$$

where $\mu$ and $\lambda$ are weight parameters. FUGAL extracts a non-convex correlation term from this objective; contrarily, we preserve convexity and thus derive a spectral guarantee:

**Theorem 3.** *The function in Equation (7) is convex for $\lambda \le \frac{(\lambda_i(\tilde{A}) - \lambda_j(\tilde{B}))^2}{2}$, for all $i, j \in \{1, 2, \ldots, n\}$, where $\lambda_i(\tilde{A})$ and $\lambda_j(\tilde{B})$ represent the eigenvalues of $\tilde{A}$ and $\tilde{B}$, respectively.*

For $\lambda = 0$, the problem in Equation (7) is convex. To derive a quasi-permutation matrix, we solve Equation (7) with $\lambda = 0$ using Adam and refine the solution by gradually increasing $\lambda$, until it diverges. This regularizer, which drives the double-stochastic matrix to a permutation matrix drastically enhances approximation accuracy, as we show in Appendix C.9.

### 3.3 M-ADAM DETAILS

Algorithm 1 outlines our Modified Adam (M-ADAM) algorithm, which initializes $P$ as an identity matrix and $\lambda$ to 0 (Line 1), and gradually increases $\lambda$ (Line 12). For each $\lambda$, it starts from the solution of the previous round and iteratively updates it using the objective's gradient (Lines 6–9). We employ the *penalty method* (Yeniay, 2005) to enforce doubly-stochastic matrix constraints. For a given value of $\lambda$, the relaxed solution $P$ is rounded to a permutation matrix $H$ via Hungarian, which is then used to transform the problem in the subsequent iteration (see § 3.4). Figure G illustrates the process with an example. M-ADAM outputs a permutation matrix that yields an edit path for the approximated GED (Kuhn, 1955). As the true GED is the least edit cost over all alignments, the

---

**Algorithm 1** M-ADAM

**Notations:**
$f = \frac{||\tilde{A}P - P\tilde{B}||_F^2}{2} + \mu \cdot (tr(P^T D)), \quad g = tr(P^T(J - P))$
$pnlt = ||P\mathbf{1} - \mathbf{1}||^2 + ||P^T\mathbf{1} - \mathbf{1}||^2 + ||max(0, -P)||^2 + ||max(0, P - J)||^2$
**Input:** matrices $\tilde{A}, \tilde{B}, D$ **Output:** permutation matrix $P$

**Algorithm:**

1:   $P \leftarrow I, \sigma \leftarrow 5, \lambda \leftarrow 0, m_0 \leftarrow 0, v_0 \leftarrow 0, \beta_1 \leftarrow 0.9, \beta_2 \leftarrow 0.99, \tilde{P} \leftarrow I, \tilde{H} \leftarrow I$
2:   **while** *true* **do**
3:     $t \leftarrow 0$
4:     **while** *not converged* **do**
5:       $t \leftarrow t + 1$
6:       $grad \leftarrow \nabla f + \sigma \cdot \nabla pnlt + \lambda \cdot \nabla g$
7:       $m_t \leftarrow \beta_1 \cdot m_{t-1} + (1 - \beta_1) \cdot grad; \hat{m}_t \leftarrow m_t/(1 - \beta_1^t)$
8:       $v_t \leftarrow \beta_2 \cdot v_{t-1} + (1 - \beta_2) \cdot grad^2; \hat{v}_t \leftarrow v_t/(1 - \beta_2^t)$
9:       $P \leftarrow P - \alpha \cdot \hat{m}_t/(\sqrt{\hat{v}_t} + \epsilon)$
10:    **if** *diverged* **then**
11:      break
12:    $\sigma \leftarrow \sigma * 2, \quad \lambda \leftarrow \lambda + 0.5$
13:    $H \leftarrow \text{Hungarian}(P), \tilde{A} \leftarrow H\tilde{A}H^\top, D \leftarrow HD$
14:    $\tilde{P} \leftarrow \tilde{H}^\top P, \tilde{H} \leftarrow H\tilde{H}$
15:    **if** $\sigma > \sigma_{th}$ **then**
16:      break
17: **return** $\tilde{P}$

---

returned GED upper-bounds the true GED. Moreover, M-ADAM is a *deterministic* algorithm; for any given pair of input matrices, it always returns the same output.

### 3.4 INVERSE RELABELING

Here, we propose an *inverse relabeling* strategy in M-ADAM. The core term of our objective is $||\tilde{A} - P\tilde{B}P^T||_F^2$, to be minimized over $\mathbb{W}^n$. After the first gradient-based update iteration with fixed $\lambda$ (outer loop in M-ADAM), we begin enforcing permutation constraints via a regularizer. Let $H$ denote a permutation matrix obtained by rounding the relaxed solution $P$ using Hungarian projection.

Since the feasible set $\mathbb{P}^n$ is discrete, gradients are computed in the relaxed domain $\mathbb{W}^n$. However, continuing the optimization near a non-identity permutation $H$ is inefficient. A non-identity $H$ acts as a rotation of the problem's coordinate system, causing the components of the gradient to become highly coupled. This motivates recentering the problem after each outer iteration. Specifically, we transform $\tilde{A} \leftarrow H\tilde{A}H^\top$. This transformation is equivalent to the variable change $\tilde{P} = H^\top P$, as:

$$\|\tilde{A} - P\tilde{B}P^\top\|_F^2 \rightarrow \|H\tilde{A}H^\top - P\tilde{B}P^\top\|_F^2 = \|\tilde{A} - H^\top P\tilde{B}\tilde{P}^\top H\|_F^2 = \|\tilde{A} - \tilde{P}\tilde{B}\tilde{P}^\top\|_F^2,$$

This variable change to $\tilde{P}$ and multiplication by $H^\top$ revokes the permutation, or *inverts the labeling*, introduced by $H$, without altering the feasible space: $\tilde{P} \in \mathbb{W}^n \iff P = H\tilde{P} \in \mathbb{W}^n$, since multiplying a doubly stochastic matrix by a permutation matrix preserves row and column sums and non-negativity. The updated $\tilde{P}$ satisfies $\tilde{P} \approx H^\top H = I$, hence gradient updates are performed in a coordinate system centered around the identity matrix $I$, allowing for more efficient and accurate corrections to small errors. Our ablation study in § C.9 validates the effectiveness of this transformation.

## 4 EXPERIMENTS

Here, we present a comprehensive evaluation of EUGENE, addressing the following aspects:

- **Efficacy:** EUGENE tops supervised and heuristic methods across datasets and costs.

- **Scalability:** EUGENE scales well to large graphs, consistently surpassing baselines.

- **Efficiency:** EUGENE incurs lower computational costs than heuristic methods with better performance; as it runs on CPUs, it curtails carbon emissions.

## 4.1 EXPERIMENTAL SETUP

Appendix C.1 outlines the hardware and software[2] environment, Appendices C.3 presents the parameters used, and Appendix C.9 reports on an ablation study.

**Baselines:** We compare EUGENE to 15 state-of-the-art supervised and optimization based heuristic methods. These include the following supervised methods: GRAPHEDX (Jain et al., 2024), GMN-EMBED (Li et al., 2019), GREED (Ranjan et al., 2022), ERIC (Zhuo & Tan, 2022), SIMGNN (Bai et al., 2019), H2MN (Zhang et al., 2021), EGSC (Qin et al., 2021), GOTSIM (Doan et al., 2021), GEDGNN (Piao et al., 2023), GMSM (Pellizzoni et al., 2024). We exclude the neural approximation algorithms GRAPHSIM (Bai et al., 2020) as GRAPHEDX and H2MN have shown vastly better performance (Jain et al., 2024; Zhang et al., 2021). Genn-A* (Wang et al., 2021) does not scale for graphs of sizes more than 10, hence excluded from the analysis. Among the neural methods included, GEDGNN, GMSM and GOTSIM provide a node mapping corresponding to the estimated GED. With all baselines, when edit costs are uniform, we use the official author-released codebases with the original training protocols and default hyperparameters. However, existing baselines do not support non-uniform edit costs, except for GRAPHEDX, which extended support to non-uniform costs and released adapted codebases for all baselines. In the non-uniform cost setting, we use these fine-tuned and publicly available versions provided by the GRAPHEDX authors.

In the heuristic methods category, we compare with the five best-performing methods from the benchmarking study by (Blumenthal et al., 2019b), namely, BRANCH-TIGHT (Blumenthal & Gamper, 2018), F1 (Lerouge et al., 2017), ADJ-IP (Justice & Hero, 2006), IPFP (Bougleux et al., 2017) and COMPACT-MIP (Blumenthal & Gamper, 2020). All these heuristic methods furnish an edit path that corresponds to the approximated GED. We utilized the GEDLIB (Blumenthal et al., 2019b) implementation of these methods in our evaluations.

**Datasets:** Table 1 lists the datasets we use. App. C.2 discusses the semantics. AIDS, Molhiv, Mutag, Code2 are labeled whereas IMDB, COIL-DEL, Triangles, Netscience and HighSchool are unlabeled.

**Train-Val-Test Splits:** As in (Jain et al., 2024), we remove isomorphic graphs from the datasets prior to training neural methods to mitigate isomorphism bias via leakage between training and testing Ivanov et al. (2019). Further, for each dataset, we restrict to the graphs of size less than 25 to ensure feasibility of ground truth GED computation. As in (Ranjan et al., 2022) and (Jain et al., 2024), we used MIP-F2 (Lerouge et al., 2017) with

Table 1: Datasets.

| Name | Avg $|\mathcal{V}|$ | Avg $|\mathcal{E}|$ | # labels | Domain |
|------|------|------|------|------|
| AIDS | 11.83 | 24.14 | 38 | Biology |
| Molhiv | 15.47 | 31.86 | 119 | Biology |
| Mutag | 23.32 | 44.64 | 14 | Biology |
| Code2 | 18.61 | 37.42 | 97 | Software |
| IMDB | 11.49 | 63.74 | - | Movies |
| COIL-DEL | 8.70 | 34.44 | - | Vision |
| Triangles | 9.11 | 20.16 | - | Synthetic |
| Netscience | 379 | 914 | - | Collaboration |
| HighSchool | 327 | 5818 | - | Proximity |

a time limit of 600 seconds for each graph pair and kept pairs that yielded equal lower and upper bounds as ground truth GED. The training set consists of $5k$ randomly sampled graph pairs, while the validation and test sets each consist of $1k$ randomly sampled pairs each.

**Cost Settings:** We evaluate the performance under three different edit cost settings:

- **Case 1 (Nonuniform costs):** The node insertion cost is 3, node deletion cost is 1, edge insertion and deletion costs are 2, and the node substitution cost is 0.

- **Case 2 (Nonuniform costs with substitution):** In addition to Case 1, substituting nodes with unequal labels incurs cost. If the substituted node label is the nearest neighbor based on the similarity ranking of node labels, the cost is 1, otherwise 2. As an illustrative case, the distance between labels is taken as the difference between their label IDs.

- **Case 3 (Uniform costs):** Node/edge insertion and deletion costs 1, node substitution 0.

Cost Settings 1 and 3 closely follow those proposed in GRAPHEDX. We introduce Cost Setting 2 to further increase the difficulty of the task. Unlike the other settings, the cost of an edit operation in this case is non-static, it dynamically varies based on the node labels involved, thereby requiring models to account for contextual variations during alignment. We also evaluate on edits costs inspired from chemistry. The results are discussed in App. C.13.

---

[2]Our C++ code and datasets are at `https://anonymous.4open.science/r/Eugene-1107/`

**Metrics:** We use two metrics to assess GED approximation and interpretability: (i) Mean Absolute Error (MAE), and (ii) Strict Interpretability (SI). MAE serves as a metric to quantify the closeness of the predicted GED to the true GED. SI is measured as the fraction of graph pairs for which the predicted GED matches the true GED. A match between the predicted and true GED indicates that the alignment produced by the method is optimal. Consequently, SI reflects the algorithm's ability to produce the optimal node mapping and serves as a measure of interpretability.

## 4.2 BENCHMARKING ACCURACY (MAE)

Table 2 presents approximation accuracy in terms of MAE on benchmark datasets under the non-uniform cost setting (Case 1) and the non-uniform cost with substitution setting (Case 2). Appendix C.4 shows the comparison under the uniform cost setting and Appendix C.5 shows that on unlabeled datasets. In all cases, EUGENE outperforms all baselines.

**Comparison with Supervised Baselines:** EUGENE outperforms all supervised baselines—including those providing node alignments—across datasets and cost settings by a large margin. Under the nonuniform cost setting, it achieves up to 44% lower MAE on Code2 and a 72% reduction on AIDS compared to the next best method. For nonuniform costs with substitution, the improvement margin ranges from 44% on Mutag to 63% on Molhiv. GRAPHEDX, EGSC, and ERIC demonstrate the second-best performance.

**Comparison with Heuristic Baselines:** EUGENE demonstrates a substantial improvement over heuristic baselines. The margin of improvement exceeds 80% across all datasets and both cost settings when compared to the next-best method, ADJ-IP. Methods BRANCH-TIGHT and COMPACT-MIP perform considerably worse than EUGENE.

Table 2 further reveals that heuristic baselines fall short of supervised ones, which explains why the community shifted to supervised methods, despite their lack of interpretability, poor generalizability, and costly training. Though heuristic, EUGENE tops supervised baselines and grants interpretability. Contrarily, supervised methods that yield node alignments tend to lag, as they trade accuracy for interpretability. EUGENE makes no such compromise.

Table 2: Accuracy comparison among baselines in MAE under different cost settings; green and yellow cells denote the best and second-best performance, respectively, for each dataset.

| Methods | Cost Setting Case 1 | | | | Cost Setting Case 2 | | | |
|---|---|---|---|---|---|---|---|---|
| | AIDS | Molhiv | Code2 | Mutag | AIDS | Molhiv | Code2 | Mutag |
| ERIC | 1.17 | 1.38 | 1.48 | 4.80 | 1.25 | 1.59 | 1.71 | 1.89 |
| EGSC | 1.35 | 1.58 | 1.65 | 1.59 | 1.35 | 1.71 | 1.79 | 1.80 |
| GRAPHEDX | 1.54 | 1.36 | 1.33 | 2.39 | 2.06 | 2.10 | 1.56 | 2.80 |
| H2MN | 1.53 | 2.00 | 1.90 | 1.74 | 1.58 | 2.08 | 2.34 | 2.00 |
| GMN-EMBED | 3.35 | 5.25 | 2.68 | 5.52 | 3.64 | 5.83 | 2.67 | 6.34 |
| GREED | 2.98 | 5.03 | 2.48 | 5.12 | 3.39 | 5.36 | 2.62 | 5.32 |
| SIMGNN | 1.55 | 1.98 | 1.85 | 1.91 | 1.70 | 2.09 | 2.01 | 2.49 |
| GEDGNN | 2.37 | 4.23 | 2.61 | 2.46 | 2.28 | 3.60 | 3.36 | 3.86 |
| GOTSIM | 7.53 | 14.49 | 8.15 | 10.89 | 10.66 | 22.19 | 12.07 | 15.38 |
| GMSM | 15.04 | 25.57 | 21.16 | 26.81 | 21.08 | 34.12 | 32.49 | 35.59 |
| BRANCH-TIGHT | 7.97 | 9.86 | 13.91 | 15.02 | 6.95 | 9.95 | 21.47 | 13.62 |
| ADJ-IP | 1.69 | 4.06 | 5.05 | 4.30 | 3.58 | 5.97 | 6.70 | 6.85 |
| F1 | 5.41 | 10.63 | 6.28 | 10.64 | 5.8 | 13.47 | 11.08 | 13.82 |
| COMPACT-MIP | 2.95 | 7.21 | 8.39 | 7.13 | 6.18 | 10.29 | 12.72 | 10.78 |
| IPFP | 5.63 | 9.99 | 6.39 | 9.53 | 8.47 | 14.27 | 13.43 | 14.36 |
| EUGENE | 0.33 | 0.65 | 0.75 | 0.68 | 0.58 | 0.79 | 0.58 | 1.01 |

**Unlabeled datasets:** We observed a similar trend on unlabeled data, as shown in App C.5, EUGENE achieving an even greater margin of improvement. That is expected, as the absence of node features limits the effectiveness of GNN-based methods, which distinguish nodes by features. We note the highest improvement with IMDB dataset, which is also the densest. High density causes oversquashing in GNNs (Giovanni et al., 2024), and is a likely reason for subpar performance of neural models.

## 4.3 ACCURACY (SI)

Table 3 presents the comparison of EUGENE with other baselines in terms of the Strict Interpretability (SI) metric. While few neural baselines do not explicitly provide alignments, we found the SI score for all supervised methods to be 0 across all cost settings. This finding indicates that, albeit some

neural methods provide explicit node alignments, they fall short in alignment quality. We thus omit these scores from the table. EUGENE consistently achieves higher SI scores compared to other heuristic methods, with an improvement of up to 69% on the Code2 dataset under cost setting Case 1. These superior SI scores highlight EUGENE's ability to deliver optimal node alignments. Although supervised baselines generally provide better GED approximations than heuristic methods, heuristic baselines offer better interpretability. EUGENE surpasses all baselines in both approximation accuracy and interpretability metrics, establishing itself as the new state-of-the-art for GED approximation while maintaining interpretability of the approximated GED.

Table 3: Accuracy comparison in terms of SI under different cost settings; green and yellow cells denote the best and second-best performance, respectively, for each dataset.

| Methods | Cost Setting Case 1 | | | | Cost Setting Case 2 | | | |
|---|---|---|---|---|---|---|---|---|
| | AIDS | Molhiv | Code2 | Mutag | AIDS | Molhiv | Code2 | Mutag |
| BRANCH-TIGHT | 0.02 | 0.02 | 0.01 | 0.01 | 0.01 | 0.01 | 0.01 | 0.01 |
| ADJ-IP | 0.90 | 0.69 | 0.48 | 0.62 | 0.69 | 0.65 | 0.63 | 0.46 |
| F1 | 0.44 | 0.10 | 0.05 | 0.04 | 0.57 | 0.15 | 0.03 | 0.07 |
| COMPACT-MIP | 0.72 | 0.31 | 0.03 | 0.20 | 0.46 | 0.31 | 0.16 | 0.24 |
| IPFP | 0.04 | 0.02 | 0.03 | 0.02 | 0.01 | 0.01 | 0.01 | 0.01 |
| EUGENE | 0.91 | 0.84 | 0.82 | 0.83 | 0.71 | 0.67 | 0.74 | 0.59 |

## 4.4 ACCURACY ON LARGE GRAPHS

The complexity of GED estimation rises with graph size due to the exponential growth of mappings in combinatorial space. We evaluate performance exclusively on large graphs to explicitly investigate this aspect of scalability. We consider graphs with sizes in the range $[25, 50]$ in the test split. Table 4 presents the MAE results under Case 1 and Case 2 cost settings, which demonstrate the superior scalability of EUGENE to large graphs, with up to 66% lower MAE than the next best performer, H2MN. Other methods exhibit significantly higher MAE. These findings underscore the practical applicability of EUGENE for GED approximation on large graphs. SI comparison on large graphs appears in Appendix C.6.

Table 4: Accuracy among baselines in MAE under different cost settings; graph sizes in $[25, 50]$; green and yellow cells denote best and second-best performance, respectively.

| Methods | Cost Setting Case 1 | | | | Cost Setting Case 2 | | | |
|---|---|---|---|---|---|---|---|---|
| | AIDS | Molhiv | Code2 | Mutag | AIDS | Molhiv | Code2 | Mutag |
| ERIC | 19.70 | 9.08 | 12.24 | 14.64 | 18.46 | 14.08 | 29.14 | 9.47 |
| EGSC | 35.68 | 12.68 | 15.02 | 15.12 | 30.22 | 16.92 | 16.04 | 14.31 |
| GRAPHEDX | 24.44 | 21.65 | 33.01 | 21.82 | 20.75 | 17.01 | 34.01 | 15.98 |
| H2MN | 6.48 | 4.59 | 5.70 | 3.44 | 10.86 | 5.15 | 10.42 | 4.54 |
| GMN-EMBED | 9.60 | 10.82 | 8.52 | 9.80 | 9.99 | 13.68 | 14.57 | 11.03 |
| GREED | 10.05 | 10.20 | 8.46 | 9.28 | 9.66 | 9.50 | 12.09 | 9.92 |
| SIMGNN | 28.77 | 10.58 | 14.02 | 7.52 | 25.61 | 12.63 | 50.51 | 12.70 |
| GEDGNN | 25.78 | 11.83 | 36.75 | 19.96 | 23.29 | 15.27 | 25.17 | 17.18 |
| GOTSIM | 29.03 | 25.93 | 26.87 | 24.62 | 29.78 | 32.47 | 31.58 | 30.48 |
| GMSM | 44.66 | 44.62 | 49.65 | 44.22 | 21.08 | 50.90 | 66.06 | 55.94 |
| BRANCH-TIGHT | 29.76 | 24.95 | 31.54 | 27.86 | 26.62 | 23.23 | 26.27 | 28.72 |
| ADJ-IP | 23.00 | 21.98 | 34.52 | 21.54 | 17.81 | 11.95 | 46.42 | 17.00 |
| F1 | 23.22 | 11.19 | 21.92 | 15.05 | 30.32 | 11.56 | 42.86 | 17.95 |
| COMPACT-MIP | 73.30 | 40.02 | 76.71 | 56.84 | 59.33 | 28.95 | 47.20 | 41.18 |
| IPFP | 17.86 | 14.65 | 16.51 | 16.48 | 18.65 | 18.47 | 24.88 | 20.16 |
| EUGENE | 4.45 | 3.88 | 4.14 | 2.80 | 3.25 | 3.73 | 4.33 | 4.74 |

Figure 2 presents MAE heatmaps on Code2 for cost setting Case 1. Each point stands for a graph pair $\mathcal{G}_Q, \mathcal{G}_T$ with coordinates (GED($\mathcal{G}_Q, \mathcal{G}_T$), ($|\mathcal{V}_Q| + |\mathcal{V}_T|$)/2). Heatmaps for EGSC, H2MN, and GRAPHEDX have a discernibly darker hue, corroborating that EUGENE enjoys better scalability in graph size and GED value. Appendix C.11 shows heatmaps for other datasets, while Appendix C.12 presents results on two thousand-scale collaboration networks, Netscience (Newman, 2006) and HighSchool (Fournet & Barrat, 2014). To our knowledge, no prior GED estimation method handles graphs of this scale.

## 4.5 COMPARISON WITH FUGAL

FUGAL addresses unrestricted graph alignment (UGA), while EUGENE estimates GED and produces an alignment corresponding to the approximation. As Theorem 1 shows, UGA is a special case

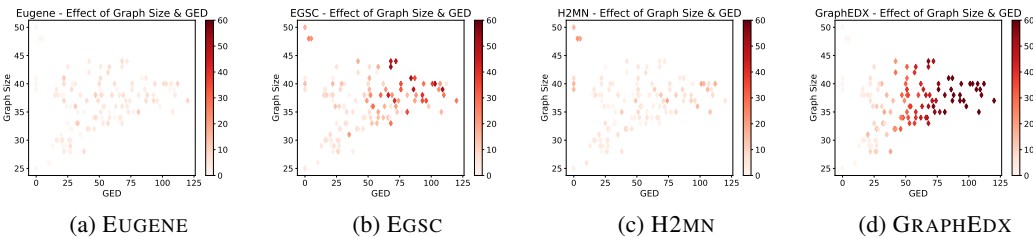

|  |  |  |  |
|:---:|:---:|:---:|:---:|
| (a) EUGENE | (b) EGSC | (c) H2MN | (d) GRAPHEDX |

Figure 2: MAE heatmap vs. graph size & GED for Code2 for graphs of size $[25, 50]$.

of GED with node edit costs set to zero. The connection between UGA and GED established in Theorem 2 allows us to draw from UGA methods, though our optimization differs in key ways:

**Optimization.** EUGENE employs a modified Adam optimizer with a penalty method to enforce doubly stochastic constraints, whereas UGA methods typically use Frank-Wolfe (Frank & Wolfe, 1956) with Sinkhorn-Knopp normalization (Cuturi, 2013). As shown in Table Q, replacing Adam with Frank–Wolfe (EUGENE-FW) leads to weaker performance, confirming the effectiveness of our approach. Our novel *inverse relabelling* strategy further improves GED estimation (§ C.9).

Table 5: GED estimation error (MAE) under Cost Setting 1.

| Method | AIDS | Molhiv | Code2 | Mutag |
|---|---|---|---|---|
| FUGAL | 7.12 | 11.72 | 6.53 | 11.60 |
| FUGAL-Node Edit Costs | 7.71 | 12.65 | 7.93 | 12.49 |
| EUGENE | **0.33** | **0.65** | **0.75** | **0.68** |

**Cost Regularizer.** EUGENE integrates node edit costs through a matrix $D$, while UGA methods may only use similar terms as structural regularizers. To test whether FUGAL could benefit from node edit costs, we evaluated it with EUGENE's cost matrix $D$. Table 5 shows that both FUGAL variants yield substantially higher GED error than EUGENE.

One might still believe that FUGAL is inherently tailored for GED instances with zero node edit costs, corresponding to UGA. We thus set all node edit costs to 0 and edge edit costs to 1. Even under this UGA-compatible setting, EUGENE demonstrated superior performance, as shown in Table 6.

Table 6: GED estimation error (MAE) under zero node edit costs (UGA setting).

| Method | AIDS | Molhiv | Code2 | Mutag |
|---|---|---|---|---|
| FUGAL | 4.71 | 6.98 | 8.52 | 8.44 |
| EUGENE | **0.28** | **0.50** | **0.74** | **0.55** |

This raises the question of why the poor GED estimates from UGA methods are not evident in UGA studies. The key difference lies in evaluation: GED is evaluated strictly by edge and node differences from the ground truth (the QAP objective), while UGA is evaluated more loosely

Table 7: Replacing EUGENE 's Frobenius norm with FUGAL's non-convex correlation term.

| Method | AIDS | Molhiv | Code2 | Mutag |
|---|---|---|---|---|
| EUGENE (FUGAL QAP) | 5.43 | 7.53 | 17.82 | 12.65 |
| EUGENE | **0.33** | **0.65** | **0.75** | **0.68** |

by the fraction of correctly aligned nodes. Hence, GED methods must enforce much stricter fidelity to the QAP objective than UGA methods, as we discuss in the following.

**Core Objective Term.** EUGENE prioritizes the convex Frobenius norm $\|AP - PB\|_F^2$, which ensures stable updates. UGA methods instead optimize the non-convex correlation term $\mathrm{Tr}(APB^\top P^\top)$ for efficiency, paired with Frank-Wolfe. Substituting this non-convex term into EUGENE caused divergence; even the best result within a 10-minute cap (Table 7) remained far less accurate. This confirms that FUGAL's core objective is ill-suited for GED estimation.

## 5 CONCLUSIONS

We introduced EUGENE, an optimization based heuristic method that provides explainable estimates of GED based on a structure-aware representation and relaxation of the underlying optimization problem. Through extensive experimentation, we demonstrated that EUGENE achieves state-of-the-art GED estimates and superior scalability compared to baselines across diverse datasets, even while it eliminates the need to generate supervisory data via **NP**-hard computations. These features position EUGENE as a promising candidate for practical graph similarity measurement. As our implementation relies solely on CPU resources, it is open to further enhancement.

## 6 REPRODUCIBILITY STATEMENT

We have made the implementation of EUGENE publicly available; the code link is provided at the end of Page 6. The released implementation includes the benchmark test sets, as well as the training and validation sets used for the neural models. We also provide scripts to generate new test sets for independent evaluation. Details on data generation, testing setup, and baseline implementations are described in Section 4. Appendix C.1 specifies the hardware and software environment, and Appendix C.3 lists the parameters used by EUGENE.

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

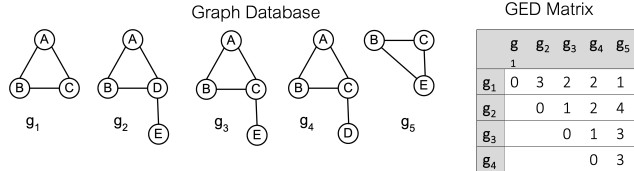

Figure C: GED among five graphs; all edit operations cost 1.

# 7 APPENDIX

## A RELATED WORK

**Supervised Methods:** GRAPHEDX (Jain et al., 2024) represents each graph as a set of node and edge embeddings and learn the alignments using a Gumbel-Sinkhorn permutation generator, additionally ensuring that the node and edge alignments are consistent with each other. GREED (Ranjan et al., 2022) employs a siamese network to generate graph embeddings in parallel and estimates the Graph Edit Distance (GED) by computing the norm of their difference. ERIC (Zhuo & Tan, 2022) eliminates the need for explicit node alignment by leveraging a regularizer and computes similarity using a Neural Tensor Network (NTN) and a Multi-Layer Perceptron (MLP) applied to graph-level embeddings obtained from a Graph Isomorphism Network (GIN). GMN (Li et al., 2019) assess graph similarity using Euclidean distance between embeddings and exist in two variants: GMN-EMBED (late interaction) and GMN-MATCH (early interaction), both utilizing message passing to capture structural similarities. SIMGNN (Bai et al., 2019) combines graph-level and node-level embeddings, where a Neural Tensor Network processes graph-level embeddings, while a histogram-based feature vector derived from node similarities enhances the similarity computation. H2MN (Zhang et al., 2021) utilizes hypergraphs to model higher-order node similarity, employing a subgraph matching module at each convolution step before aggregating the final graph embeddings via a readout function and passing them through an MLP. EGSC (Qin et al., 2021) introduces an Embedding Fusion Network (EFN) within a Graph Isomorphism Network (GIN) to generate unified embeddings for graph pairs, which are further processed through an EFN and an MLP to compute the final similarity score. GOTSIM (Doan et al., 2021) approximates GED through a neural network, and simultaneously learns the alignments. Specifically, it formulates the similarity between a pair of graphs as the minimal "transformation" cost from one graph to another in the learnable node-embedding space. GEDGNN (Piao et al., 2023) treats GED computation as a regression task and predict the GED value. A post-processing algorithm based on $k$-best matching is used to extract node mapping. GMSM (Pellizzoni et al., 2024) uses regularized optimal transport with GNNs to approximate GED.

**Heuristic Methods:** F1 (Lerouge et al., 2017), ADJ-IP (Justice & Hero, 2006), and COMPACT-MIP (Blumenthal & Gamper, 2020) employ a mixed integer programming framework based on the LP-GED paradigm to approximate the GED. In contrast, BRANCH-TIGHT (Blumenthal & Gamper, 2018) iteratively solves instances of the linear sum assignment problem or the minimum-cost perfect bipartite matching problem. IPFP (Bougleux et al., 2017) models GED as a quadratic assignment problem and uses Integer Projected Fixed Point method to aproximate the QAP.

## B PROOFS

**Theorem 1.** *Given graphs $\mathcal{G}_1$ and $\mathcal{G}_2$ of size $n$, if the edge insertion and deletion cost is $\kappa^2 = 2$ and node substitution cost is 0, then $GED(\mathcal{G}_1, \mathcal{G}_2) = \min_{\pi \in \Phi(\mathcal{G}_1, \mathcal{G}_2)} \|AP_\pi - P_\pi B\|_F^2$.*

*Proof.* We derive the set of edge insertions and deletions to convert $\mathcal{G}_1$ to $\mathcal{G}_2$ from $\pi$. An edge that should be inserted between nodes $i$ and $j$ in $\mathcal{G}_1$ does not exist in $A$ but exists in $B$, hence $a_{ij} = 0$ and $b_{\pi(i)\pi(j)} = 1$. Likewise, an edge that needs deletion has $a_{ij} = 1$ and $b_{\pi(i)\pi(j)} = 0$. All other $(i, j)$ pairs have $a_{ij} = b_{\pi(i)\pi(j)}$. Let $\mathcal{E}_{ins}$ be the set of edges to be inserted in $\mathcal{G}_1$ and $\mathcal{E}_{del}$ that of edges to be deleted by $\pi$, where without loss of generality an edge $(i, j)$ has $i < j$. As node edit

and edge substitutions cost 0, the $GED_\pi(\mathcal{G}_1, \mathcal{G}_2)$ with respect to edit operations induced by $\pi$ is:

$$GED_\pi(\mathcal{G}_1, \mathcal{G}_2) = \sum_{(i,j)\in\mathcal{E}_{ins}} 2 + \sum_{(i,j)\in\mathcal{E}_{del}} 2 = \sum_{(i,j)\in\mathcal{E}_{ins}} 2\cdot(a_{ij}-b_{\pi(i)\pi(j)})^2 + \sum_{(i,j)\in\mathcal{E}_{del}} 2\cdot(a_{ij}-b_{\pi(i)\pi(j)})^2$$

$$= \sum_{(i,j)\in\mathcal{E}_{ins}} \left((a_{ij}-b_{\pi(i)\pi(j)})^2 + (a_{ji}-b_{\pi(j)\pi(i)})^2\right) + \sum_{(i,j)\in\mathcal{E}_{del}} \left((a_{ij}-b_{\pi(i)\pi(j)})^2 + (a_{ji}-b_{\pi(j)\pi(i)})^2\right)$$

$$+ \sum_{i<j,(i,j)\notin\mathcal{E}_{del}\cup\mathcal{E}_{ins}} \left((a_{ij}-b_{\pi(i)\pi(j)})^2 + (a_{ji}-b_{\pi(j)\pi(i)})^2\right)$$

$$= \sum_{(i,j)\in[n]\times[n]} (a_{ij}-b_{\pi(i)\pi(j)})^2 = \|A - P_\pi B P_\pi^T\|_F^2 = \|A P_\pi - P_\pi B\|_F^2$$

By the given edit costs, $GED(\mathcal{G}_1, \mathcal{G}_2) = \min_\pi \{GED_\pi(\mathcal{G}_1, \mathcal{G}_2)\}$, hence,

$$GED(\mathcal{G}_1, \mathcal{G}_2) = \min_{\pi\in\Phi(\mathcal{G}_1,\mathcal{G}_2)} \|A P_\pi - P_\pi B\|_F^2$$

$\square$

**Theorem 2.** *Given two graphs $\mathcal{G}_1$ and $\mathcal{G}_2$ of size $n$, $GED(\mathcal{G}_1, \mathcal{G}_2) = \min_{\pi\in\Phi(\mathcal{G}_1,\mathcal{G}_2)} \frac{\|\tilde{A} P_\pi - P_\pi \tilde{B}\|_F^2}{2} + tr(P_\pi^T D)$, where $\tilde{A}$, $\tilde{B}$ and $D$ are defined as above.*

*Proof.* We first reformulate Equation 4 as follows:

$$\frac{\|\tilde{A} P_\pi - P_\pi \tilde{B}\|_F^2}{2} + tr(P_\pi^T D) = \frac{\|\tilde{A} - P_\pi \tilde{B} P_\pi^T\|_F^2}{2} + tr(P_\pi^T D)$$

Using the node-alignment function $\pi$, we reformulate the above to:

$$\sum_{(i,j)\in[n]\times[n]} \kappa^2 \cdot \frac{(a_{ij}-b_{\pi(i)\pi(j)})^2}{2} + \sum_{i\in[n]} d_{i,\pi(i)}$$

Further manipulation via the definition of matrix $D$ gives:

$$\sum_{(i,j)\in[n]\times[n]} \kappa^2 \cdot \frac{(a_{ij}-b_{\pi(i)\pi(j)})^2}{2} + \sum_{i\in\mathcal{G}_1\text{ is a dummy}} d_v(\epsilon, \mathcal{L}(\pi(i))) + \sum_{i\in\mathcal{G}_1\text{ mapped to dummy }\pi(i)} d_v(\mathcal{L}(i), \epsilon) + \sum_{\mathcal{L}(i)\neq\mathcal{L}(\pi(i))} d_v(\mathcal{L}(i), \mathcal{L}(\pi(i)))$$

Notably, for any $(i,j) \in [n]\times[n]$, if $a_{ij}=0$ and $b_{\pi(i)\pi(j)}=1$, an $(i,j)$ edge should be inserted. Likewise, if $a_{ij}=1$ and $b_{\pi(i)\pi(j)}=0$, edge $(i,j)$ should be deleted. Otherwise, if $a_{ij}=b_{\pi(i)\pi(j)}$, the term evaluates to 0. Besides, a dummy node $i$ in $\mathcal{G}_1$ should be inserted with $\pi(i)$ as the corresponding node in $\mathcal{G}_2$, while a node $i$ mapped to a dummy node $\pi(i)$ should be deleted. In the event that none of these conditions apply, node $i$ is substituted with node $\pi(i)$. We thus simplify the expression to:

$$\sum_{(i,j)\text{ inserted}} \kappa^2 \cdot \frac{b_{\pi(i)\pi(j)}^2 + b_{\pi(j)\pi(i)}^2}{2} + \sum_{(i,j)\text{ deleted}} \kappa^2 \cdot \frac{a_{ij}^2 + a_{ji}^2}{2} + \sum_{i\in\mathcal{G}_1\text{ is inserted}} d_v(\epsilon, \mathcal{L}(\pi(i))) +$$

$$\sum_{i\in\mathcal{G}_1\text{ is deleted}} d_v(\mathcal{L}(i), \epsilon) + \sum_{i\in\mathcal{G}_1\text{ is replaced with }\pi(i)} d_v(\mathcal{L}(i), \mathcal{L}(\pi(i)))$$

Substituting the values, we obtain:

$$\sum_{(i,j)\text{ inserted}} \kappa^2 + \sum_{(i,j)\text{ deleted}} \kappa^2 + \sum_{i\in\mathcal{G}_1\text{ is inserted}} d_v(\epsilon, \mathcal{L}(\pi(i))) + \sum_{i\in\mathcal{G}_1\text{ is deleted}} d_v(\mathcal{L}(i), \epsilon) + \sum_{i\in\mathcal{G}_1\text{ is replaced with }\pi(i)} d_v(\mathcal{L}(i), \mathcal{L}(\pi(i)))$$

$$= GED_\pi(\mathcal{G}_1, \mathcal{G}_2) \tag{8}$$

Since $GED(\mathcal{G}_1, \mathcal{G}_2) = \min_\pi \{GED_\pi(\mathcal{G}_1, \mathcal{G}_2)\}$ and $\min_{\pi\in\Phi(\mathcal{G}_1,\mathcal{G}_2)} \frac{\|\tilde{A} P_\pi - P_\pi \tilde{B}\|_F^2}{2} + tr(P_\pi^T D) = GED_\pi(\mathcal{G}_1, \mathcal{G}_2)$, $GED(\mathcal{G}_1, \mathcal{G}_2) = \min_{\pi\in\Phi(\mathcal{G}_1,\mathcal{G}_2)} \frac{\|\tilde{A} P_\pi - P_\pi \tilde{B}\|_F^2}{2} + tr(P_\pi^T D)$. $\square$

**Lemma 1.** *A doubly-stochastic matrix $A$ with $tr(A^T(J-A)) = 0$ is a permutation matrix.*

*Proof.* Given that $tr(A^T(J - A)) = 0$, it follows that $\sum_i \sum_j a_{ij} \cdot (1 - a_{ij}) = 0$. Since $A$ is doubly-stochastic, $0 \leq a_{ij} \leq 1$ for all $i$ and $j$, hence $a_{ij} \cdot (1 - a_{ij})$ is non-negative for $1 \leq i, j \leq n$. Thus, $a_{ij} \cdot (1 - a_{ij}) = 0$ for all $i$ and $j$. It follows that $a_{ij}$ must be either 0 or 1 for each $i$ and $j$. As $A$ is doubly-stochastic and all its entries are either 0 or 1, by definition $A$ is a permutation matrix. $\square$

**Theorem 3.** *The function in Equation* (7) *is convex for* $\lambda \leq \frac{(\lambda_i(\tilde{A}) - \lambda_j(\tilde{B}))^2}{2}$, *for all* $i, j \in \{1, 2, \ldots, n\}$, *where* $\lambda_i(\tilde{A})$ *and* $\lambda_j(\tilde{B})$ *represent the eigenvalues of* $\tilde{A}$ *and* $\tilde{B}$, *respectively.*

*Proof.* We begin by considering the first term in Equation (7), $\frac{1}{2} \|\tilde{A}P - P\tilde{B}\|^2$. The second derivative of this term is given by: $I \otimes \tilde{A}^2 - 2 \cdot (\tilde{B} \otimes \tilde{A}) + \tilde{B}^2 \otimes I$, where $\otimes$ denotes the Kronecker product, and $I$ represents the identity matrix. The second term in the equation is linear in the matrix $P$, implying that its second derivative is zero. The second derivative of the third term is given by: $-2\lambda(I \otimes I)$. Thus, the Hessian matrix of the entire function is:

$$I \otimes \tilde{A}^2 - 2 \cdot (\tilde{B} \otimes \tilde{A}) + \tilde{B}^2 \otimes I - 2\lambda(I \otimes I).$$

For the function to be convex, the Hessian must be positive semidefinite, which requires that its eigenvalues be non-negative. This leads to the condition:

$$\lambda \leq \frac{\lambda_i(\tilde{A})^2 + \lambda_j(\tilde{B})^2 - 2\lambda_i(\tilde{A})\lambda_j(\tilde{B})}{2} = \frac{(\lambda_i(\tilde{A}) - \lambda_j(\tilde{B}))^2}{2}, \tag{9}$$

for all $i, j \in \{1, 2, \ldots, n\}$, where $\lambda_i(\tilde{A})$ and $\lambda_j(\tilde{B})$ are the eigenvalues of matrices $\tilde{A}$ and $\tilde{B}$, respectively. $\square$

## C  EXPERIMENTS

### C.1  HARDWARE AND SOFTWARE ENVIRONMENTS

We ran all experiments on a machine equipped with an Intel Xeon Gold 6142 CPU @1GHz and a GeForce GTX 1080 Ti GPU. While heuristic methods including EUGENE run on the CPU, supervised baselines exploit the GPU.

### C.2  DATASETS

The semantics of the datasets are as follows:
- **AIDS** (Morris et al., 2020): A compilation of graphs originating from the AIDS antiviral screen database, representing chemical compound structures.
- **OGBG-Molhiv** (Molhiv) (Hu et al., 2020): Chemical compound datasets of various sizes, where each graph represents a molecule. Nodes correspond to atoms, and edges represent chemical bonds. The atomic number of each atom serves as the node label.
- **OGBG-Code2** (Code2) (Hu et al., 2020): A collection of Abstract Syntax Trees (ASTs) derived from approximately 450,000 Python method definitions. Each node in the AST is assigned a label from a set of 97 labels. We considered the graphs as undirected.
- **Mutagenicity** (Mutag) (Debnath et al., 1991): A chemical compound dataset of drugs categorized into two classes: mutagenic and non-mutagenic.
- **IMDB** (Yanardag & Vishwanathan, 2015): This dataset consists of ego-networks of actors and actresses who have appeared together in films. The graphs in this dataset are unlabelled.
- **COIL-DEL** (Riesen & Bunke, 2008): This dataset comprises graphs extracted from images of various objects using the Harris corner detection algorithm. The resulting graphs are unlabelled.
- **Triangles** (Knyazev et al., 2019): This is a synthetically generated dataset designed for the task of counting triangles within graphs. The graphs in this dataset are unlabelled.

### C.3  PARAMETERS

Table H lists the parameters used for EUGENE. We set the convergence criterion of M-ADAM to $abs(prev\_dist - cur\_dist) < 1e^{-7}$, where $prev\_dist$, $cur\_dist$ are the approximated Graph edit distances in two successive iterations, $itr - 1$ and $itr$.

Table H: Parameters used in EUGENE.

| parameter | value |
|---|---|
| $\mu$ | 1 |
| $\alpha$ | 0.001 |
| $\sigma_{th}$ | $1e^3$ |

Table I: Accuracy Comparison among baselines for unit edit costs. Cells shaded in green denote the best performance in each dataset.

| Methods | MAE | | | | SI | | | |
|---|---|---|---|---|---|---|---|---|
| | AIDS | Molhiv | Code2 | Mutag | AIDS | Molhiv | Code2 | Mutag |
| ERIC | 0.57 | 0.66 | 0.56 | 0.65 | 0.00 | 0.00 | 0.00 | 0.00 |
| EGSC | 0.70 | 0.81 | 0.80 | 0.82 | 0.00 | 0.00 | 0.00 | 0.00 |
| GRAPHEDX | 0.65 | 0.85 | 0.59 | 0.78 | 0.00 | 0.00 | 0.00 | 0.00 |
| H2MN | 0.86 | 0.94 | 0.84 | 0.89 | 0.00 | 0.00 | 0.00 | 0.00 |
| GMN-EMBED | 0.61 | 0.75 | 0.76 | 1.15 | 0.00 | 0.00 | 0.00 | 0.00 |
| GREED | 0.59 | 0.82 | 0.75 | 0.75 | 0.00 | 0.00 | 0.00 | 0.00 |
| SIMGNN | 0.77 | 0.90 | 0.79 | 1.06 | 0.00 | 0.00 | 0.00 | 0.00 |
| GEDGNN | 1.19 | 2.16 | 1.50 | 1.89 | 0.00 | 0.00 | 0.00 | 0.00 |
| GOTSIM | 3.36 | 5.20 | 9.76 | 4.74 | 0.00 | 0.00 | 0.00 | 0.00 |
| GMSM | 7.34 | 13.04 | 10.01 | 13.32 | 0.00 | 0.00 | 0.00 | 0.00 |
| BRANCH-TIGHT | 4.13 | 4.98 | 6.79 | 7.05 | 0.02 | 0.02 | 0.02 | 0.01 |
| ADJ-IP | 0.45 | 2.16 | 2.32 | 2.27 | 0.83 | 0.69 | 0.50 | 0.62 |
| F1 | 2.6 | 5.48 | 2.82 | 5.39 | 0.48 | 0.13 | 0.14 | 0.05 |
| COMPACT-MIP | 1.49 | 4.17 | 3.93 | 4.07 | 0.75 | 0.27 | 0.01 | 0.18 |
| IPFP | 2.81 | 5.19 | 2.85 | 4.97 | 0.08 | 0.02 | 0.14 | 0.02 |
| EUGENE | 0.26 | 0.55 | 0.72 | 0.58 | 0.87 | 0.74 | 0.69 | 0.72 |

## C.4 ACCURACY UNDER UNIFORM EDIT COST SETTING

Table I presents the approximation accuracy results in terms of MAE and SI on benchmark datasets under the uniform cost setting (Case 3). For MAE, EUGENE outperforms all baselines on the AIDS, Molhiv, and Mutag datasets, while on the Code2 dataset, ERIC outperforms EUGENE. In terms of SI, EUGENE consistently surpasses all considered baselines. These results establish EUGENE as a robust method capable of accurately estimating GED across diverse cost settings. The difficulty (i.e., MAE) increases as costs become more diverse (i.e., from uniform to non-uniform costs) and the size of the considered edit space expands (i.e., from zero to non-zero cost of substitution). We thus observe the lowest MAE in Setting 3, followed by Setting 1, and the highest MAE in Setting 2.

## C.5 ACCURACY ON UNLABELLED DATASETS

Table J presents the accuracy comparison of IMDB, COIL-DEL, and Triangles datasets in terms of MAE for cost setting Case 1 and Case 3. As these datasets are unlabelled, Case 2 is not applicable. EUGENE consistently outperforms both supervised and heuristic baselines across all scenarios, demonstrating its robustness and effectiveness for GED prediction across diverse datasets.

## C.6 SI ON LARGE GRAPHS

Table 3 presents a comparison of EUGENE with other baselines in terms of the Strict Interpretability (SI) metric for graphs of sizes $[25, 50]$. EUGENE consistently achieves significantly higher SI scores compared to other heuristic methods. These superior SI scores on large graphs highlight EUGENE's enhanced scalability in delivering interpretable GED, outperforming other non-neural methods.

## C.7 CARBON EMISSIONS

Table L presents the total carbon emissions for the top-performing models across various datasets. EUGENE was executed on a CPU, which operates at a power consumption of approximately 150 watts

Table J: Accuracy Comparison among baselines in terms of MAE under different cost settings for unlabelled datasets. Cells shaded in greendenote the best performance in each dataset.

| Methods | Cost Setting Case 1 | | | Cost Setting Case 3 | | |
|---|---|---|---|---|---|---|
| | IMDB | COIL-DEL | Triangles | IMDB | COIL-DEL | Triangles |
| ERIC | 10.42 | 1.41 | 2.65 | 3.80 | 1.87 | 1.47 |
| EGSC | 5.96 | 3.23 | 3.80 | 6.50 | 3.89 | 2.82 |
| GRAPHEDX | 7.10 | 1.41 | 2.26 | 1.46 | 1.21 | 0.50 |
| H2MN | 15.51 | 8.44 | 7.02 | 7.20 | 4.27 | 3.38 |
| GMN-EMBED | 4.75 | 2.93 | 3.41 | 1.37 | 0.89 | 0.63 |
| GREED | 5.02 | 2.90 | 3.39 | 1.39 | 0.88 | 0.73 |
| SIMGNN | 7.58 | 2.00 | 2.36 | 3.73 | 1.04 | 0.97 |
| GEDGNN | 10.78 | 3.54 | 1.97 | 3.31 | 1.69 | 1.16 |
| GOTSIM | 25.01 | 9.41 | 6.94 | 8.20 | 4.19 | 2.84 |
| GMSM | 40.70 | 20.18 | 16.94 | 19.67 | 9.97 | 8.20 |
| BRANCH-TIGHT | 7.22 | 6.47 | 5.68 | 3.58 | 3.30 | 2.71 |
| ADJ-IP | 1.58 | 0.71 | 0.40 | 1.22 | 0.23 | 0.30 |
| F1 | 8.68 | 3.75 | 1.58 | 4.26 | 1.75 | 0.82 |
| COMPACT-MIP | 17.05 | 4.01 | 1.04 | 9.56 | 2.10 | 0.64 |
| IPFP | 18.87 | 8.67 | 7.04 | 9.15 | 4.27 | 3.47 |
| EUGENE | 1.02 | 0.43 | 0.21 | 0.15 | 0.21 | 0.17 |

Table K: Accuracy comparison among baselines in terms of SI under different cost settings for graphs of sizes $[25, 50]$. Cells shaded in green denote the best performance in each dataset.

| Methods | Cost Setting Case 1 | | | | Cost Setting Case 2 | | | |
|---|---|---|---|---|---|---|---|---|
| | AIDS | Molhiv | Code2 | Mutag | AIDS | Molhiv | Code2 | Mutag |
| BRANCH-TIGHT | 0.12 | 0.03 | 0.05 | 0.09 | 0.01 | 0.05 | 0.04 | 0.04 |
| ADJ-IP | 0.18 | 0.03 | 0.09 | 0.10 | 0.25 | 0.08 | 0.13 | 0.10 |
| F1 | 0.04 | 0.00 | 0.01 | 0.04 | 0.03 | 0.04 | 0.10 | 0.03 |
| COMPACT-MIP | 0.00 | 0.00 | 0.00 | 0.01 | 0.05 | 0.05 | 0.05 | 0.04 |
| IPFP | 0.01 | 0.01 | 0.00 | 0.01 | 0.00 | 0.00 | 0.01 | 0.00 |
| EUGENE | 0.35 | 0.31 | 0.30 | 0.46 | 0.36 | 0.26 | 0.18 | 0.16 |

under full load. In contrast, all other neural models utilized a GPU, which consumes approximately 250 watts under full load. Our carbon emission estimation follows a standard methodology:

$$\text{Energy Consumption} = \text{Power (kW)} \times \text{Time (hours)}$$
$$\text{CO}_2 \text{ Emissions} = \text{Energy Consumption} \times 475 \text{ gCO}_2\text{/kWh}$$

The emission factor of 475 gCO$_2$/kWh is sourced from International Energy Agency (2019). The carbon emissions account for the time taken to generate ground truth, training, and inference for the neural models, whereas EUGENE, being optimization-based, only includes inference time. While we acknowledge that training and ground-truth computation costs would be amortized over many inferences, it is reasonable to include those costs for any model that requires them. EUGENE demonstrates significantly lower carbon emissions compared to the supervised methods, achieving up to 30 times lower emissions on the Molhiv dataset.

Table L: Total Carbon Emissions (in grams of $CO_2$).

| Model | AIDS | Molhiv | Code2 | Mutag |
|---|---|---|---|---|
| ERIC | 75.56 | 204.22 | 71.42 | 222.82 |
| EGSC | 78.43 | 215.27 | 73.23 | 223.85 |
| GRAPHEDX | 410.65 | 612.09 | 426.40 | 251.55 |
| H2MN | 437.91 | 442.55 | 123.21 | 277.00 |
| EUGENE | 6.06 | 7.11 | 8.11 | 7.28 |

Table M: Running times (MM:SS) on benchmark datasets.

| Methods | AIDS | Molhiv | Code2 | Mutag | IMDB | COIL-DEL | Triangles |
|---|---|---|---|---|---|---|---|
| EUGENE | 05:06 | 05:59 | 06:50 | 06:05 | 05:09 | 05:06 | 04:59 |
| BRANCH-TIGHT | 00:24 | 00:48 | 03:52 | 01:17 | 00:18 | 00:07 | 00:13 |
| ADJ-IP | 02:38 | 06:34 | 09:33 | 07:12 | 05:42 | 02:06 | 01:29 |
| IPFP | 00:20 | 00:45 | 01:40 | 01:20 | 00:15 | 00:05 | 00:09 |
| COMPACT-MIP | 10:02 | 11:47 | 12:53 | 12:16 | 07:35 | 07:38 | 05:01 |
| F1 | 08:04 | 11:01 | 11:52 | 10:51 | 09:56 | 08:13 | 05:11 |

## C.8 EFFICIENCY

Table M presents the running time of optimization based heuristic methods on the benchmark datasets for the entire test set. Among these methods, IPFP and BRANCH-TIGHT demonstrates the fastest runtimes but exhibits the poorest accuracy among all 15 baselines in Table 2 across datasets and cost settings. Excluding BRANCH-TIGHT and IPFP, EUGENE achieves superior runtime performance compared to other optimization based methods on the Molhiv, Code2, Mutag and IMDB datasets. On the AIDS, COIL-DEL and Triangles datasets, ADJ-IP demonstrates better run times, and EUGENE is second best. Importantly, EUGENE achieves a significant accuracy advantage while maintaining competitive efficiency, reinforcing its position as both an effective and efficient solution for GED approximation.

**Time Complexity Analysis:** The objective function (Eq. (7)) includes matrix multiplications with a worst-case time complexity of $\mathcal{O}(n^3)$. Gradient calculations also have a worst-case complexity of $\mathcal{O}(n^3)$ due to matrix multiplications. Thus, the overall time complexity becomes $\mathcal{O}(T \cdot n^3)$, where $T$ is the number of computation epochs. Additionally, as the algorithm is CPU-bound, GED computations for each graph pair can be massively parallelized by leveraging multi-core CPUs and hyperthreading.

**Impact of Time Budgets:** As certain heuristic baselines employ time constraints, we retained their default parameter settings to ensure consistency. To examine how performance varies with increased computational budget, we conducted an analysis on the Code2 dataset under **Cost Setting 1**, using time budgets of 5, 10, and 15 minutes. The results are presented in Table N.

Table N: GED estimation error (MAE) on Code2 under varying time budgets (minutes).

| Method | 5 min | 10 min | 15 min |
|---|---|---|---|
| BRANCH-TIGHT | 13.91 | 13.87 | 13.88 |
| ADJ-IP | 6.98 | 5.05 | 3.96 |
| COMPACT-MIP | 24.14 | 8.40 | 6.10 |
| F1 | 16.31 | 6.28 | 7.72 |
| IPFP | 6.44 | 6.47 | 6.39 |

Branch-Tight and IPFP converged within 5 minutes, as evidenced by the absence of any improvement in MAE with larger time budgets. The remaining three methods exhibited modest gains when given additional time, suggesting that they benefit from prolonged optimization. Still, Eugene achieves a MAE of 0.75 within 7 minutes, outperforming all baselines even at the maximum allotted time.

Table O: Accuracy (MAE) of EUGENE vs. EUGENE'.

| Datasets | Cost Setting Case 1 | | Cost Setting Case 2 | |
|---|---|---|---|---|
| | EUGENE | EUGENE' | EUGENE | EUGENE' |
| AIDS | **0.33** | 10.51 | **0.58** | 9.22 |
| Molhiv | **0.65** | 9.96 | **0.79** | 11.63 |
| Code2 | **0.75** | 13.46 | **0.58** | 6.04 |
| Mutag | **0.68** | 19.12 | **1.01** | 16.10 |

Table P: Accuracy (MAE) of EUGENE vs. EUGENE-NoIR.

| Datasets | Cost Setting Case 1 | | Cost Setting Case 2 | |
|---|---|---|---|---|
| | EUGENE | EUGENE-NoIR | EUGENE | EUGENE-NoIR |
| AIDS | **0.33** | 0.80 | **0.58** | 1.15 |
| Molhiv | **0.65** | 1.16 | **0.79** | 1.57 |
| Code2 | **0.75** | 1.19 | **0.58** | 1.02 |
| Mutag | **0.68** | 1.14 | **1.01** | 1.53 |

Table Q: Accuracy comparison of EUGENE with EUGENE-FW in Cost Setting 1

| Methods | AIDS | Molhiv | Code2 | Mutag |
|---|---|---|---|---|
| EUGENE-FW | 6.67 | 11.79 | 6.59 | 13.09 |
| EUGENE | 0.33 | 0.65 | 0.75 | 0.68 |

## C.9 ABLATION STUDY

We have so far evaluated EUGENE, which refines a doubly stochastic matrix toward a quasi-permutation matrix using a permutation-inducing regularizer before rounding. For comparison, we introduce a variant, EUGENE', which *directly rounds* the doubly stochastic solution without this regularization. As shown in Table O, EUGENE yields substantially lower MAE, highlighting the benefit of guiding the solution closer to a permutation before rounding.

We also assess the impact of the inverse relabelling strategy of M-ADAM, which recenters the problem after each iteration. To this end, we define a variant, EUGENE-NoIR, that omits this transformation. Table P reports MAE for both variants: EUGENE consistently outperforms EUGENE-NoIR, demonstrating the importance of performing gradient updates in coordinates aligned with the identity.

We also investigate the effect of using the Frank-Wolfe (FW) algorithm in place of Adam within Algorithm 1. As shown in Table Q, the M-Adam variant significantly outperforms the version that employs FW (EUGENE-FW), demonstrating the effectiveness of our optimizer choice.

## C.10 PARAMETER SENSITIVITY

We analyze the sensitivity of the M-Adam algorithm to the parameters listed in Appendix C.3, as shown in Tables R and S. A lower value of $\mu$ increases the weight of edge costs, whereas a higher $\mu$ prioritizes node costs. Across all datasets, $\mu = 1$ yields the best performance. We use $\alpha = 0.001$ (the default value for Adam), which performs best on three out of four datasets.

To examine the impact of the $\lambda$-scheduling in M-ADAM, we conducted experiment where the increment step was varied, results are presented in Table T

- **Increment = 0.1:** The influence of permutation constraints remained weak throughout optimization, leading to under-constrained solutions and suboptimal performance.

- **Increment = 0.5:** This yielded the best results, striking a balance between exploration and constraint enforcement, and was adopted as the default setting in Eugene.

- **Increment = 1, 2:** The optimizer rapidly enforced hard permutation constraints, prematurely narrowing the search space and degrading solution quality.

These results emphasize the importance of a carefully tuned $\lambda$-schedule in achieving both accuracy and stability in GED estimation.

## C.11 IMPACT OF GRAPH SIZE AND GED

Section 4.4 presented heatmaps of MAE vs. graph size and true GED value on the Code2 dataset. Heatmaps for the AIDS, Molhiv, and Mutag datasets are provided in Figs. D- F. The conclusions remain consistent: GRAPHEDX, EGSC, and H2MN exhibit noticeably darker tones across the spectrum compared to EUGENE, highlighting EUGENE's superior scalability with respect to GED and graph sizes across datasets.

Table R: Accuracy comparision with varying $\mu$

| $\mu$ | AIDS | Molhiv | Code2 | Mutag |
|---|---|---|---|---|
| 0.1 | 3.07 | 9.13 | 4.97 | 5.31 |
| 0.2 | 2.29 | 7.5 | 2.96 | 3.98 |
| 0.5 | 0.9 | 3.39 | 0.9 | 1.36 |
| 1 | 0.58 | 0.79 | 0.58 | 1.01 |
| 2 | 0.85 | 1.14 | 0.84 | 1.69 |

Table S: Accuracy comparision with varying $\alpha$

| $\alpha$ | AIDS | Molhiv | Code2 | Mutag |
|---|---|---|---|---|
| 0.1 | 0.61 | 0.82 | 0.62 | 0.98 |
| 0.01 | 0.58 | 0.81 | 0.61 | 1.02 |
| 0.001 | 0.58 | 0.79 | 0.58 | 1.01 |

Table T: Effect of varying $\lambda$-increment step on GED estimation error (MAE).

| Increment step | AIDS | molhiv | code2 | Mutag |
|---|---|---|---|---|
| 0.1 | 1.45 | 2.08 | 1.40 | 2.10 |
| 0.5 | **0.33** | **0.65** | **0.75** | **0.68** |
| 1 | 0.80 | 1.54 | 2.77 | 1.85 |
| 2 | 3.19 | 6.18 | 10.01 | 9.88 |

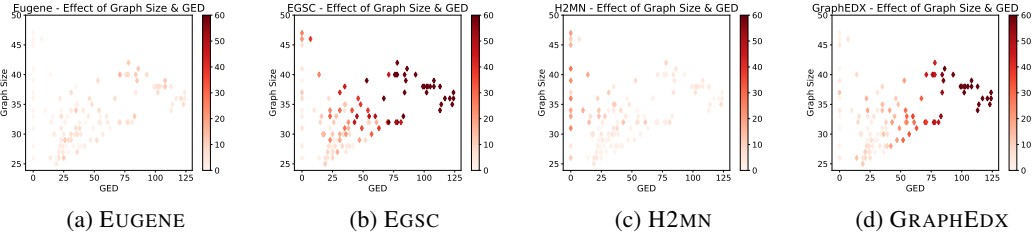

(a) EUGENE     (b) EGSC     (c) H2MN     (d) GRAPHEDX

Figure D: MAE heatmap vs. graph size & GED for AIDS for graphs of size $[25, 50]$.

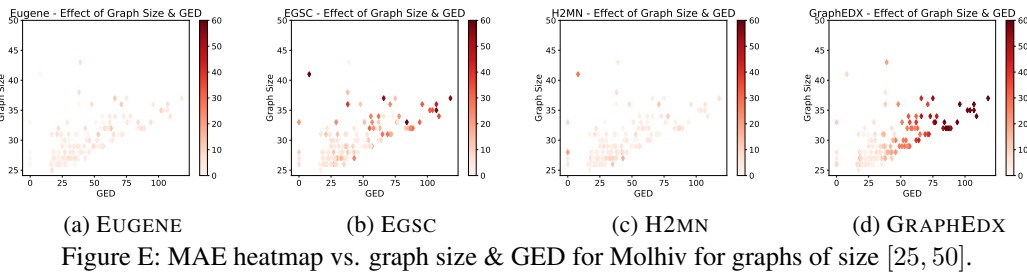

(a) EUGENE     (b) EGSC     (c) H2MN     (d) GRAPHEDX

Figure E: MAE heatmap vs. graph size & GED for Molhiv for graphs of size $[25, 50]$.

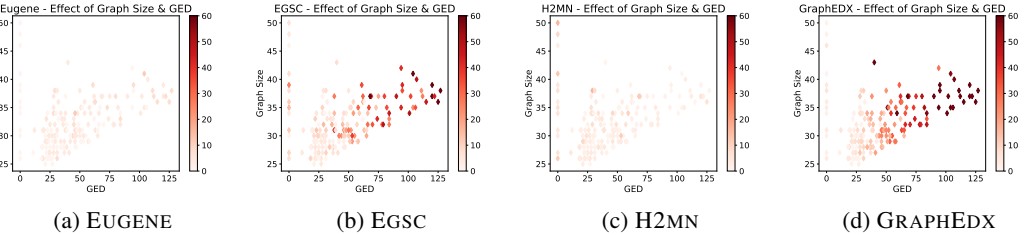

(a) EUGENE     (b) EGSC     (c) H2MN     (d) GRAPHEDX

Figure F: MAE heatmap vs. graph size & GED for Mutag for graphs of size $[25, 50]$.

## C.12 ACCURACY ON VERY LARGE GRAPHS

As detailed in (Blumenthal & Gamper, 2018), GED methods are traditionally applied to small-scale graphs due to computational complexity. We extend the feasibility of GED approximation to substantially larger graphs. We present results on two unlabelled thousand-scale collaboration network datasets, Netscience ($|V| = 379$, $|E| = 914$) and HighSchool ($|V| = 327$, $|E| = 5818$) in Table U. To our knowledge, no prior GED approximation benchmark handles graphs of this scale. On HighSchool, a dense evolving dataset, we compute the GED of the last graph version from versions containing 80%, 85%, 90%, and 99% of edges. On NetScience, we create five graphs by introducing small noise to the original graph. Since it's not feasible to create a training set with exact ground-truth GED for such large graphs, we excluded neural models from our analysis. IPFP didn't terminate within a time limit of 3 hrs. Results clearly indicate superior scalabilty of EUGENE both in terms of MAE and running times.

Table U: Performance comparison on HighSchool and NetScience Datasets

| Methods | MAE | | Running Time (sec) | |
|---|---|---|---|---|
| | HighSchool | NetScience | HighSchool | NetScience |
| ADJ-IP | 4568 | 152.99 | 2695 | 1446 |
| BRANCH-TIGHT | 582 | 833 | 1115 | 2369 |
| F1 | 5032 | 859.4 | 1912 | 1526 |
| EUGENE | 0 | 22.8 | 961 | 1372 |

## C.13 ILLUSTRATIVE EXAMPLE WITH DOMAIN-SPECIFIC EDIT COSTS

To model meaningful structural similarity, we design edit costs with domain-specific heuristics from chemistry for Molhiv dataset.

**Node Substitution Cost.** Substituting one atom for another alters a molecule's electronic properties, reactivity, and biological function. To account for these effects, node substitution costs are assigned based on the electronegativity difference between atoms:

- **Low Cost (1):** Applied when the electronegativity difference is less than $0.2$. These substitutions typically involve chemically similar atoms that frequently co-occur in analogous functional groups.
- **Moderate Cost (2):** Assigned when the difference lies in $[0.2, 0.7]$, indicating moderate chemical dissimilarity.
- **High Cost (3):** Used when the difference exceeds $0.7$, reflecting substitutions likely to disrupt molecular structure and activity.

**Node Insertion / Deletion Cost.** The cost of inserting or deleting a node is determined by the bond multiplicity of the associated atom:

- **Cost = 3:** Atom participates in at least one triple bond.
- **Cost = 2:** Atom participates in at least one double bond but no triple bond.
- **Cost = 1:** Atom is involved only in single bonds.

This hierarchy reflects the increasing structural and energetic disruption when removing atoms from more rigid bonding environments.

**Edge Insertion / Deletion Cost.** Edge insertion and deletion costs are set uniformly to **1**.

Table V: GED estimation error (MAE) on Molhiv under chemistry-informed edit costs.

| Method | EUGENE | ERIC | EGSC | GRAPHEDX | GREED |
|---|---|---|---|---|---|
| Molhiv | **1.30** | 2.09 | 1.94 | 1.74 | 2.58 |

We presented the results in Table V. Eugene outperforms competing baselines under the proposed chemistry-informed edit cost setting, demonstrating its ability to effectively capture real-world molecular similarity.

### C.14 Illustrative Example of Eugene 's Pipeline

We considered two graphs of sizes 12, 11 respectively and show four stages of EUGENE 's operation: (i) the initial mapping; (ii) the doubly stochastic matrix generated after the first iteration of Algorithm 1 ($\lambda = 0$); (iii) the quasi-permutation matrix at the end of third iteration of Algorithm 1; (iv) The final mapping returned by EUGENE. The optimal transformation from Graph 1 to Graph 2 involves removing node 10, removing the edge from node 1 to node 4, and adding an edge from node 5 to node 9 in Graph 1. As the figure shows, by the third iteration, our novel regularizer has turned the doubly stochastic matrix to a sparse one. At the end, the algorithm achieves the *optimal* node alignment. After iteration 3, nodes 6 and 7 of Graph 1 have similar weightage for nodes 5 and 6 of Graph 2, as these nodes share similar structural neighborhoods. Node 10 is mapped to node 11, which is a dummy node in Graph 2, indicating that it should be deleted.

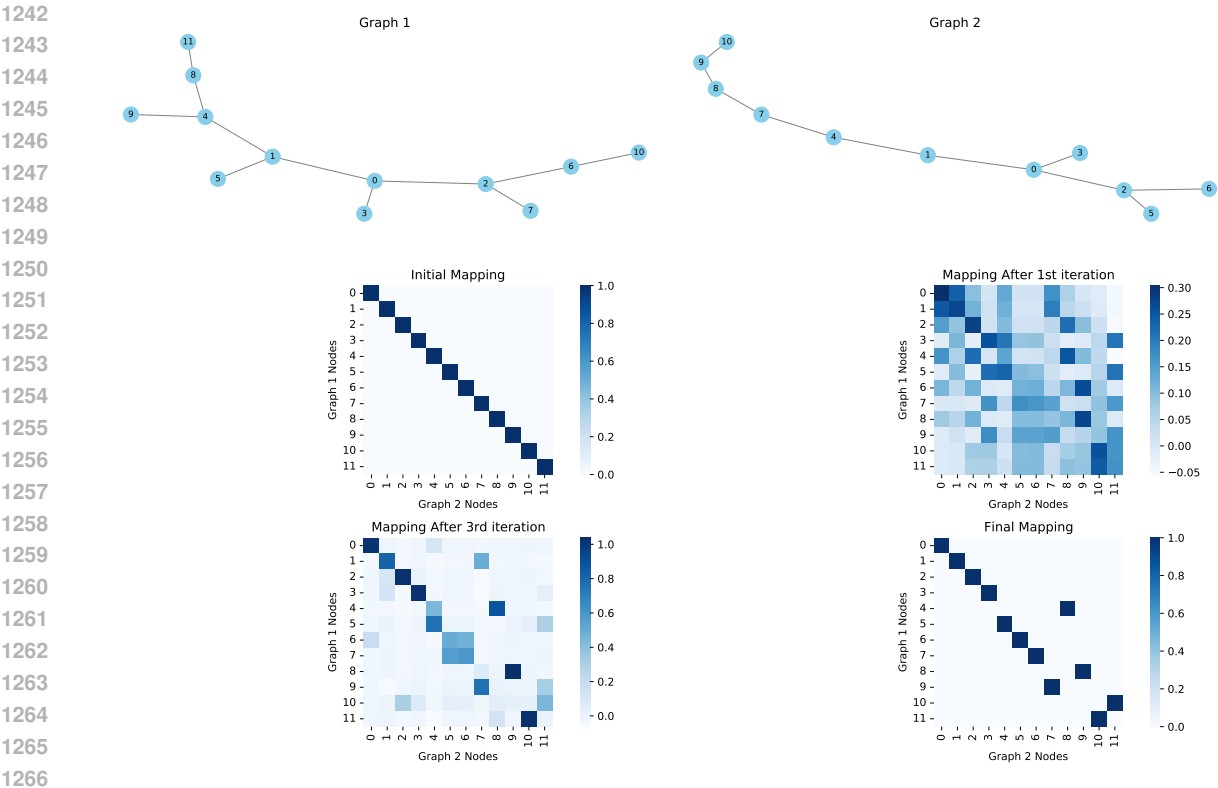

Figure G: Operational stages of EUGENE