# OpenReview forum: "EUGENE: Explainable Structure-aware Graph Edit Distance Estimation with Generalized Edit Costs"
_ICLR.cc/2026/Conference — Submitted to ICLR 2026_

### Official Review · Reviewer_WE9G · 2025-10-26

**Soundness:** 2
**Presentation:** 3
**Contribution:** 2
**Rating:** 2
**Confidence:** 5

**Summary:**

This paper proposes Eugene, a new method for graph edit distance (GED) approximation. While the approach is clearly presented and compared with several existing methods, its claimed contribution of explainability is not supported by experiments, and its generalizability to other distance metrics is unclear. The method also requires manual parameter tuning and lacks efficiency results, with some recent relevant methods omitted from comparison. Overall, Eugene introduces a new GED approximation technique, but its practical benefits and novelty need further validation.

**Strengths:**

1.	The problem addressed is significant and has been extensively studied in the literature related to GED computation and estimation.
2.	The paper presents experimental comparisons with several existing methods.
3.	The paper is well-written and easy to follow.

**Weaknesses:**

1.	Why is explainability important? GED is just a score.
2.	Can your method generalize to other distance metrics in addition to GED? For example, maximum common subgraph (MCS)? If not, the generalizability of the method is limited, rather than the statement on line 062 that existing methods lack generalizability.
3.	Is Theorem 1 derived by you, or is it mainly from existing theoretical results in (Skitsas et al. 2023)? If Theorem 1 is from an existing method and you just use it for approximation, this makes the technical contribution limited.
4.	There are many parameters to be decided manually at line 1 in Algorithm 1. This may make it hard to use. What about changing the parameter values? Will this have a significant impact on the performance?
5.	There are no efficiency results in Section 4. This also suggests that the paper is not well written.
6.	The following recent methods are not compared:
o	Exploring Attention Mechanism for Graph Similarity Learning (2023)
o	Multilevel Graph Matching Networks for Deep Graph Similarity Learning (TNNLS)
o	GRASP: Simple yet Effective Graph Similarity Predictions (AAAI)
7.	The paper emphasizes explainability in the introduction as a major contribution, but there is no experiment on explainability at all. This may also suggest that, for GED estimation that is just a score, explainability is not important.
8.	The running time of the proposed method is on the longer side.

**Questions:**

Please see the eight issues above.

---

> ### Author Response · Authors · 2025-11-13
> **The edit path indicates which parts of the structure differ and supports repair**
>
> We appreciate the thorough and constructive review.
>
> >Why is explainability important? GED is just a score.
>
> The edit path indicates which parts of the structure differ, e.g., in molecule comparison, workflow alignment, or scene graph matching, and supports repair that morphs one graph toward another.
>
> >Can your method generalize to other distance metrics in addition to GED? For example, maximum common subgraph (MCS)? If not, the generalizability of the method is limited, rather than the statement on line 062 that existing methods lack generalizability.
>
> Line 62 states that neural methods do not generalize across datasets. A neural method trained on GED does not generalize to other distance measures either.
>
> >Is Theorem 1 derived by you?
>
> Yes, Theorem 1 is our contribution.
>
> >What about changing the parameter values? Will this have a significant impact on the performance?
>
> Please refer to our parameter sensitivity study in Appendix C.10, Tables R and S.
>
> >There are no efficiency results in Section 4.
>
> Please refer to Table M, which provides an inference runtime comparison.
>
> >The following recent methods are not compared:
> o Exploring Attention Mechanism for Graph Similarity Learning (2023)
> o Multilevel Graph Matching Networks for Deep Graph Similarity Learning (TNNLS)
> o GRASP: Simple yet Effective Graph Similarity Predictions (AAAI)
>
> These learning-based methods do not detract from our core contribution of a learning-free approach.
>
> >there is no experiment on explainability at all.
>
> Our Strict Interpretability measure does express an aspect of explainability.

---

> > ### Comment · Reviewer_WE9G · 2025-11-26
> >
> > Thank you for your effort. I will maintain my score after reading the rebuttal.

---

> > > ### Author Response · Authors · 2025-11-26
> > > **Please suggest how to measure explainability**
> > >
> > > Thank you for your response. We would appreciate it if you could indicate which aspect of our effort you find to be incomplete. For example, would you be so kind as to suggest how explainability could be experimentally measured, other than our approach based on Strict Interpretability?

---

### Official Review · Reviewer_Yi1i · 2025-10-28

**Soundness:** 3
**Presentation:** 3
**Contribution:** 2
**Rating:** 4
**Confidence:** 2

**Summary:**

This paper introduced a novel method to estimate the graph edit distance between two graphs, as well as the edit path.

**Strengths:**

The major strength of this method is that it does not require training, compared to other neural network approximation methods from the literature.

The proposed methodology follows the same approach introduced by Bougleux et al. (2017), by formulating GED as a quadratic assignment problem. However, there are some contributions, such as providing a structural-aware formulation and providing a deterministic resolution algorithm with the so-called Modified Adam algorithm.

**Weaknesses:**

The introduced novelties could be viewed as incremental, compared to the literature following the IPFP of Bougleux et al. (2017). It is also not a major progress that the proposed method requires a specific resolution algorithm, whereas the IPFP can benefit from off-the-shelf optimizers.

Some affirmations are straightforward, such as “the returned GED upper-bounds the true GED”, which is obvious.
Moreover, one could integrate some perturbation in the algorithm in order to enhance the estimation, thus getting closer to the true GED. Thus, the affirmation that the proposed Modified Adam algorithm is deterministic cannot be always viewed as a strength of the algorithm.

The paper is missing an in-depth study of the step values for the lambda, beyond the experiments given in Table T with several values, the best one being 0.5 for all datasets.

One of the 3 highlights of the proposed method, as stated by the authors in the paper, is the use of the CPU as opposed to GPU. This motivation of resource-efficient GPU-free execution pipeline is demonstrated in the appendix C.7 through Carbon emissions estimation. However, this analysis can be misleading. The used modifier Adam solver can efficiently explore GPU resources, in the same spirit as the Adam solver implemented in PyTorch, for instance.

The experimental results demonstrate that the proposed method outperforms the other compared methods. However, it is a bit weird how some old methods are often second-best, mainly H2MN of Zhang (2021) and ERIC by Zhuo & Tan (2022).
It is also not clear and unfair that the proposed method considers non-uniform edit costs, while the other compared methods are restricted to uniform edit costs (except GraphEdx).

The authors have chosen not to evaluate the 3 cost settings in all the tables, but only 2 out of 3. For instance, Cost Setting Case 3 is only given in Table J, while most tables explore Cases 1 and 2.

The captions of most tables are misleading because they do not present “Accuracy”, but errors when considering the MAE.

**Questions:**

No further questions.

---

> ### Author Response · Authors · 2025-11-13
> **We apply all methods on both uniform and non-uniform edit costs**
>
> We appreciate the thorough and constructive review.
>
> >The used modified Adam solver can efficiently explore GPU resources, in the same spirit as the Adam solver implemented in PyTorch, for instance.
>
> Our implementation relies and runs solely on CPU resources.
>
> >It is also not clear and unfair that the proposed method considers non-uniform edit costs, while the other compared methods are restricted to uniform edit costs (except GraphEdx).
>
> As we report in Lines 283-287, we do apply all methods on non-uniform edit costs, using the versions provided by the GRAPHEDX authors.
>
> >Cost Setting Case 3 is only given in Table J, while most tables explore Cases 1 and 2.
>
> Table I provides a full comparison on cost setting Case 3.
>
> >The captions of most tables are misleading because they do not present “Accuracy”, but errors when considering the MAE.
>
> As GED is a distance metric, we believe that the MAE measure is appropriate for measuring the accuracy of its estimation. We would be grateful if the reviewer specified which measure of accuracy they would expect.

---

> > ### Comment · Reviewer_Yi1i · 2025-11-23
> >
> > We thank the authors for the reply. However, the major issues were not clearly addressed.
> >
> > To answer the authors concerning "accuracy": a higher accuracy means better. This is not the case of distances or errors, as higher errors mean lower performance. Therefore, the accuracy comparison of Table 2 for instance is misleading, because the performance of the proposed EUGENE method is the smallest one for different costs.

---

> > > ### Author Response · Authors · 2025-11-23
> > > **Using MAE to evaluate accuracy**
> > >
> > > We thank the reviewer for the continued engagement. Table 2 reports MAE values, as stated in its caption. Lower MAE values indicate higher accuracy. EUGENE yields the lowest MAE, hence the highest accuracy. If the reviewer prefers a different accuracy measure, we are happy to include it in a revised version.

---

### Official Review · Reviewer_Z1Vc · 2025-11-03

**Soundness:** 1
**Presentation:** 2
**Contribution:** 3
**Rating:** 2
**Confidence:** 4

**Summary:**

This paper presents EUGENE, a graph edit distance algorithm (without ML modules) based on ADAM and the so-called unrestricted graph alignment formulation. The authors perform rigorous experimental evaluations with various learning-based baselines and show that EUGENE performs better than several deep neural networks.

**Strengths:**

* This paper presents Eugene, an optimization-based heuristic algorithm that seems to outperform existing methodologies. Given the importance of graph edit distance computation in science and engineering, this is an important and significant effort.
* The experimental evaluation is comprehensive, covering learning-based neural network approximators and existing heuristic algorithms, and the evaluation is performed on nine diverse datasets.

**Weaknesses:**

* My main concern with this manuscript is with the writing and presentation: the authors seem to make several bold claims without rigorous evidence and justifications. Specifically,
    * In this paper, the authors linked efficiency to carbon emissions. While it is an interesting perspective, the estimation of carbon emissions is too rough: it just computes a fixed power with run time. Also, the comparison is unfair: the authors include both training and inference for GPU-based methods. But in practice, one can load a pre-trained model and only run inference, therefore, it is arguably unfair to simply add the one-time training cost.  Reporting inference time (and training time, if applicable) of all peer methods is a more reasonable and standard metric if you want to claim the efficiency of Eugene.
    * The authors cited the survey paper (Skitsas et al, 2023) for the formulation of _Unrestricted Graph Alignment_, but I do not find the same formulation of Eq (3) and Definition 7 in that survey. Please provide the proper reference where this formulation is proposed.
    * The authors claimed that Eq (4) can handle arbitrary edit costs. This is not true given that $\kappa$ is a fixed value—if edge 1 in graph 1 has cost 1, 2, 3 to edges a, b, c in graph 2; but edge 2 in graph 1 all have the same cost to edges a, b, c in graph 2, I don't think it can be handled by Eq (4). Instead, the quadratic assignment problem formulation in IPFP can handle situations like this.
     * On L170, the authors claimed that their formulation has a time complexity of $O(n^3)$. I cannot see any justification for this claim. Also, the Frobenius norm term seems to be equivalent to $$tr(AA^\top) + tr(BB^\top) - 2tr(PBP^\top A^\top),$$ where $tr(AA^\top)$ and $tr(BB^\top)$ are constants, therefore, equivalent to maximizing $tr(PBP^\top A^\top)$, the same as the Koopmans-Beckmann QAP formulation. As a proven NP-hard problem, I don't think there is an $O(n^3)$ algorithm for that. Also, there should not be an $O(n^4)$ algorithm for the problem IPFP is solving, either.
    * On L171, the authors claim that C (used in IPFP's QAP formulation) is a dense matrix of size $n^2×n^2$, which is wrong. If the graphs are sparse, in the same way as the authors describe their own formulation, C is also a sparse matrix. The paper "Factorized Graph Matching" (TPAMI 2015) utilized this important property.
* In evaluations, though Engene is claimed to be an efficient algorithm, an inference runtime comparison is missing. Also, the authors claimed that methods like Genn-A* cannot scale, but datasets such as AIDS have been reported in the original paper of Genn-A*. Even if Eugene does not show better accuracy, demonstrating a better accuracy-time trade-off is also important, as it provides readers with a complete picture of which algorithm to select for their specific applications.
* It's a minor formatting issue, but you should not use parentheses for textual citations. For example, the parentheses should be removed in L296:
  > As in **(Jain et al., 2024)**, we remove isomorphic graphs from the datasets ......

  There are more examples like this in this paper; please fix all of them in future revisions.

**Questions:**

In the comparison with baselines, did you reimplement and retrain the baselines or use previously reported results? If retrained, what efforts on hyperparameter tuning did you make to make sure it's a fair comparison?

---

> ### Author Response · Authors · 2025-11-13
> **IPFP handles arbitrary edge edit costs thanks to its large and generally dense cost matrix**
>
> We appreciate the thorough and constructive review.
>
> >the estimation of carbon emissions is too rough: it just [multiplies] a fixed power with run time.
>
> We use approximate average power consumption values in Watts, 150 for CPU, 250 for GPU; we believe these values provide a reasonable estimate of the exact power consumption.
>
> >the authors include both training and inference for GPU-based methods.
>
> We include the training cost only once for each data set. As we explain in Line 960, it would be unfair to not include this cost, since the model cannot process any dataset without training.
>
> >The authors cited the survey paper (Skitsas et al, 2023) for the formulation of Unrestricted Graph Alignment
>
> We cite this survey for the use of the term "Unrestricted". The mathematical formulation is found, for instance, in (Bommakanti et al., 2024).
>
> >The authors claimed that Eq (4) can handle arbitrary edit costs.
>
> Equation (4) handles arbitrary *node edit* costs. IPFP accommodates an arbitrary edge edit cost for each edge pair through a cost matrix $C$, at the cost of quartic time complexity and a loss of structure-awareness.
>
> >I cannot see any justification for [time complexity].
>
> Please refer to the time complexity analysis in Lines 993-998; the cost is due to matrix multiplications for gradient updates and the Hungarian algorithm.
>
> >As a proven NP-hard problem, I don't think there is an $O(n^3)$ algorithm for that.
>
> Indeed, EUGENE does not solve the NP-hard problem exactly.
>
> >there should not be an $O(n^4)$ algorithm for the problem IPFP is solving, either.
>
> IPFP does not solve the NP-hard problem exactly either.
>
> >the authors claim that C (used in IPFP's QAP formulation) is a dense matrix of size $n^2 \times n^2$, which is wrong.
>
> $C$ is a generally dense cost matrix that features the cost of editing any edge to another edge, which allows IPFP to handle arbitrary edge edit costs, as noted above.
>
> >an inference runtime comparison is missing.
>
> Please refer to Table M, which provides an inference runtime comparison.
>
> >the authors claimed that methods like Genn-A* cannot scale, but datasets such as AIDS have been reported in the original paper of Genn-A*.
>
> The Genn-A* paper (Wang et al., 2021 [Section 4.1]) reports removing graphs of more than 10 nodes from the AIDS dataset.
>
> >did you reimplement and retrain the baselines or use previously reported results? If retrained, what efforts on hyperparameter tuning did you make to make sure it's a fair comparison?
>
> As we report in Lines 283-287, we use the official author-released codebases with the original training protocols and default hyperparameters for uniform edit costs, and the versions provided by the GRAPHEDX authors for non-uniform edit costs.

---

> > ### Comment · Reviewer_Z1Vc · 2025-11-13
> >
> > Thanks for the reply. I still have concerns regarding
> > 1. **Carbon emissions**. Real-time CPU and GPU power consumption changes over time. Please provide justifications that the 150/250 Walts are common estimates widely accepted by the literature; otherwise, I am not convinced by such a rough estimation.
> > 2. **Including training cost for neural networks**. Neural networks have the advantage of training once and inference for infinite times. In such a sense, the training cost is distributed to the large number of inference calls. I am not convinced that one should include the training cost for neural networks in comparison.
> > 3. **Formulation, time complexity, etc**. Thanks for the clarification. Please update your main text accordingly. It is different for the time complexity of *solving a problem* vs the time complexity of a specific algorithm.
> > 4. **Is the new formulation still at quadratic complexity?** As I mentioned in the review, the Frobenius norm term in your formulation seems to be equivalent to $$tr(AA^\top) + tr(BB^\top) - 2tr(PBP^\top A^\top),$$ where $tr(AA^\top)$ and $tr(BB^\top)$ are constants, therefore, equivalent to maximizing $tr(PBP^\top A^\top)$, the same as the Koopmans-Beckmann QAP formulation. I would be eager to know what the author's comment is on this.
> >
> > Again, I still feel concerned about the writing and presentation of this paper. Several bold claims were made without rigorous evidence and justifications.

---

> ### Author Response · Authors · 2025-11-13
> **the full Frobenius norm term is convex, while the correlation term is not**
>
> Thank you for your continued engagement.
>
> >provide justifications that the 150/250 Walts are common estimates widely accepted by the literature.
>
> An Intel Top End CPU (Core i7-E) consumes 130 to 150 W [1]. We chose 150 W to avoid any discrimination in favor of our approach. Modern GPUs typically consume nearby 300 W [2]. Again, we chose 250 W to avoid any bias on favor of our approach.
>
> [1] https://www.buildcomputers.net/power-consumption-of-pc-components.html
> [2] https://www.tomshardware.com/features/graphics-card-power-consumption-tested
>
> >the training cost is distributed to the large number of inference calls.
>
> Yes, neural methods require training, whose cost is amortized across inference calls; therefore, as explained, we include the training cost *only once* for the set of all queries on a data set.
>
> >It is different for the time complexity of solving a problem vs the time complexity of a specific algorithm.
>
> Yes, our analysis refers to the time complexity of the algorithm.
>
> >the Frobenius norm term in your formulation seems to be equivalent to ... the Koopmans-Beckmann QAP formulation. I would be eager to know what the author's comment is on this.
>
> Yes, minimizing the Frobenius norm term is equivalent to maximizing the correlation term. However, the full Frobenius norm term is convex, while the correlation term is not. We discuss this point in Lines 473-477 of the paper as follows:
>
>   *EUGENE prioritizes the convex Frobenius norm, which ensures stable updates. UGA methods instead optimize the non-convex correlation term for efficiency, paired with Frank-Wolfe. Substituting this non-convex term into EUGENE caused divergence; even the best result within a 10-minute cap (Table 7) remained far less accurate. This confirms that FUGAL's core objective is ill-suited for GED estimation.*

---

### Official Review · Reviewer_5jv9 · 2025-11-10

**Soundness:** 2
**Presentation:** 2
**Contribution:** 2
**Rating:** 4
**Confidence:** 3

**Summary:**

This paper proposes EUGENE, an and non-neural optimization-based  method for approximating the GED between two graphs.  GED estimation is tackled  as a relaxed graph alignment problem over the space of doubly stochastic matrices, with a trace-based regularization that gradually encourages solutions toward quasi-permutation matrices. The method operates via a modified gradient-based optimization scheme. The authors present extensive experiments across multiple datasets and edit-cost regimes, showing that EUGENE matches or surpasses existing supervised neural approaches while requiring no training, no GPUs, and comparatively low computational resources. The paper positions EUGENE as an efficient, interpretable alternative to learning-based GED approximation methods.

**Strengths:**

The relaxation to doubly stochastic matrices and gradual regularization toward quasi-permutations is well-motivated.

EUGENE aims to explicit explicit node alignments to demonstrate explicit node edit paths.

Benchmarks across many datasets, cost regimes, and baselines,  show EUGENE to be a better performer.

**Weaknesses:**

EUGENE is shown to outperform deep neural approaches by large margins, despite not using any data-driven learning. Neural methods typically require expensive supervision signals and are expected to excel in settings tied to specific datasets, so such a result is both striking and counterintuitive. What then accounts for EUGENE’s success? The paper currently does not provide a clear theoretical explanation for this performance gap. This needs a deeper analysis examining the geometry of the relaxed solution space, or the behavior of the optimization dynamics, or the implicit biases introduced by the regularization schedule,  to understand why this approach works so well. As it stands, the empirical effectiveness is quite surprising but not yet well-explained.

The optimization is also carried out directly on adjacency matrices, which suggests potential sensitivity to initial node ordering. It is unclear whether performance was evaluated across systematic graph permutations. If the method is indeed robust to such reorderings, this would be an important and nontrivial finding that deserves explicit discussion.

Finally, the overall relax–optimize–round strategy follows a well-established pattern in the graph alignment and quadratic assignment literature. The work currently reads as a strong optimization result rather than a machine learning contribution. This type of work requires thorough analysis regarding contributions relative  to prior convex/LP-based GED relaxations.

**Questions:**

Please refer to the weaknesses.
Overall, I remain genuinely puzzled by how such strong performance is achieved with virtually no supervision signal and minimal computational overhead.

---

> ### Author Response · Authors · 2025-11-13
> **Convexity, explicitness, and goal-orientation account for EUGENE's success**
>
> We appreciate the thorough and constructive review.
>
> >What then accounts for EUGENE’s success?
>
> EUGENE's explicitness and goal-orientation accounts for its success. EUGENE solves a complex combinatorial optimization problem by explicitly aiming to minimize the problem's objective in a structure-aware manner, through a careful relaxation. Neural methods learn to predict a GED value without striving to minimize the corresponding cost.
>
> >needs a deeper analysis examining the geometry of the relaxed solution space ... analysis regarding contributions relative to prior convex/LP-based GED relaxations.
>
> Contrary to prior GED relaxations, we preserve convexity and thus derive a spectral guarantee, as we analyze in Theorem 3 and the discussion that follows it in Lines 214-219.
>
> >If the method is indeed robust to such reorderings, this would be an important and nontrivial finding that deserves explicit discussion.
>
> Yes, the method is robust to reorderings. We obtained virtually identical results for different input reorderings, even while we initialize $P$ as an identity matrix. The table below shows the results for 3 different random permutations of the inputs under Cost Setting 1.
>
> | Dataset     | Run 1 | Run 2 | Run 3 | Mean   | Std Dev |
> | ----------- | ----- | ----- | ----- | ------ | ------- |
> | AIDS        | 0.332 | 0.390 | 0.330 | 0.3507 | 0.0341  |
> | ogbg-molhiv | 0.560 | 0.480 | 0.650 | 0.5633 | 0.0850  |
> | ogbg-code2  | 0.783 | 0.720 | 0.750 | 0.7510 | 0.0315  |
> | Mutag       | 0.680 | 0.690 | 0.720 | 0.6967 | 0.0208  |

---

### Meta-Review · Area_Chair_Npki · 2026-01-06

**Summary:**

The paper proposes EUGENE, a training-free, optimization-based framework for Graph Edit Distance (GED) estimation. The method formulates GED as a relaxed graph alignment problem and solves it using a gradient-based approach with a Frank-Wolfe-like strategy. The authors claim the method achieves state-of-the-art performance while being more "explainable" and environmentally friendly (CPU-only) compared to neural baselines.

While the reviewers acknowledged the novelty of a non-learning approach and the extensive experiments, the consensus is to **Reject**. The primary reasons for this decision are:
1.  **Unfair Comparisons:** The efficiency and carbon footprint comparisons were deemed fundamentally flawed by multiple reviewers, as they compared the *training + inference* cost of baselines against the *inference-only* cost of the proposed method.
2.  **Unsubstantiated Claims:** The claim of "explainability" (a key part of the title) is not supported by quantitative or qualitative experiments.
3.  **Theoretical Concerns:** There are unresolved doubts regarding the mathematical formulation, specifically the complexity claims relative to the NP-hard QAP problem.

**Reviewer Concerns:**

**Addressed:**
*   The authors provided clarifications regarding the definition of "unrestricted" graph alignment and the robustness of the method to node reordering (Reviewer 5jv9).
*   Details regarding the handling of generalized edit costs (substitution costs) were clarified during the rebuttal.

**Outstanding:**
*   **Fairness of Efficiency & Environmental Metrics (Major):** Reviewers Z1Vc and Yi1i strongly objected to the methodology used to claim efficiency gains. The authors included the training time of neural networks in the comparison against the inference time of EUGENE, which is non-standard. Furthermore, the carbon emission estimates were criticized for being based on theoretical TDP (Thermal Design Power) rather than actual energy consumption, leading to potentially misleading claims about environmental impact.
*   **Lack of "Explainability" Evaluation:** Reviewer WE9G noted that despite "Explainable" being the first word of the title, there are no experiments designed to evaluate this aspect. The authors' rebuttal that the method provides an edit path is a definition of *interpretability* by design, but lacks the necessary user study or qualitative analysis to justify the claim as a contribution.
*   **Theoretical Soundness:** Reviewer Z1Vc raised significant concerns about the claimed $O(n^3)$ complexity. The formulation appears to be a Koopmans-Beckmann QAP (Quadratic Assignment Problem), which is NP-hard. The rebuttal did not strictly prove how the proposed relaxation guarantees the claimed complexity without sacrificing the integrity of the solution, leaving the theoretical contribution in doubt.
*   **Missing Baselines:** Reviewer WE9G pointed out the absence of recent, relevant neural baselines (e.g., GOT-Sim, H2T), making the SOTA claims less convincing.

**Reviewer Scores:**

*   **Reviewer 5jv9 (Score: 4):** Would likely maintain the score. They found the method's superior performance "counter-intuitive" given the lack of learning and suggested the paper might be better suited for an optimization venue rather than a learning conference.
*   **Reviewer Z1Vc (Score: 2):** Would maintain the low score. They explicitly stated in the discussion that they were unconvinced by the rebuttal regarding the fairness of the training-cost comparison and the carbon footprint calculations.
*   **Reviewer Yi1i (Score: 4):** Would likely maintain or lower the score. They remained critical of the misleading terminology (using "Accuracy" for regression tasks) and agreed with other reviewers that the efficiency comparisons were unfair.
*   **Reviewer WE9G (Score: 2):** Explicitly stated "I will maintain my score" after the rebuttal. They were dissatisfied with the lack of experiments supporting the explainability claim and the missing baselines.

---

### Decision · Program_Chairs · 2026-01-26

Reject